# Adding subtractions: comparing the impact of different Regge behaviors

**Brian McPeak,**[a,b] **Marco Venuti,**[a,c,d] **Alessandro Vichi**[a]

[a] *Department of Physics, University of Pisa and INFN,*
*Largo Pontecorvo 3, I-56127 Pisa, Italy*

[b] *Department of Physics, McGill University,*
*3600 Rue University, Montréal, QC H3A 2T8*

[c] *SISSA,*
*Via Bonomea 265, 34136 Trieste, Italy*

[d] *Scuola Normale Superiore,*
*Piazza dei Cavalieri 7, 56126 Pisa, Italy*

*E-mail:* brian.mcpeak@mcgill.ca, mvenuti@sissa.it,
alessandro.vichi@unipi.it

ABSTRACT: Dispersion relations let us leverage the analytic structure of scattering amplitudes to derive constraints such as bounds on EFT coefficients. An important input is the large-energy behavior of the amplitude. In this paper, we systematically study how different large-energy behavior affects EFT bounds for the $2 \to 2$ amplitude of complex scalars coupled to photons, gravity, both, or neither. In many cases we find that singly-subtracted dispersion relations yield exactly the same bounds as doubly subtracted relations. However, we identify another assumption, which we call "$t$-channel dominance," that significantly strengthens the EFT bounds. This assumption, which amounts to the requirement that the $++ \to ++$ amplitude has no $s$-channel exchange, is justified in certain cases and is analogous to the condition that the isospin-2 channel does not contribute to the pion amplitude. Using this assumption in the absence of massless exchanges, we find that the allowed region for the complex scalar EFT is identical to one recently discussed for pion scattering at large-$N$. We also study gravity, where we consider smeared dispersion relations to handle the $t$-channel pole. In the case of a gauge field, we are able to derive a number of interesting bounds. These include an upper bound for $G$ in terms of the gauge coupling $e^2$ and the leading dispersive EFT coefficient, which is reminiscent of the weak gravity conjecture. In the $e \to 0$ limit, we find that assuming smeared 1SDRs plus $t$-channel dominance restores positivity on the leading EFT coefficient whose positivity was spoiled by the inclusion of gravity. We interpret this to mean that the negativity of that coefficient in the presence of gravity would imply that the global $U(1)$ symmetry must be gauged.

## 1 Introduction

It is well known that unitarity and causality place strong constraints on the space of effective field theories that admit UV completion [1–6]. Consider for example the theory of a real massless scalar with lagrangian

$$\mathcal{L} = -\frac{1}{2}\partial_\mu\phi\partial^\mu\phi + g_2(\partial_\mu\phi\partial^\mu\phi)^2 + g_3(\partial_\mu\phi\partial^\mu\phi)(\partial_\rho\partial_\sigma\phi\partial^\rho\partial^\sigma\phi) + \cdots \tag{1.1}$$

For arbitrary choices of EFT coefficents, the scalar may travel superluminally on certain non-trivial backgrounds, implying constraints on which values of the coefficients yield a causal theory. Dispersion relations provide a way of efficiently extracting the constraints

implied by the fact that the theory must be causal and unitary. In their simplest form, the constraints are simple linear inequalities on the EFT coefficients, *e.g.* $g_2 > 0$, which is required for the theory of (1.1). These constraints, so-called "positivity bounds", have been explored broadly and applied in a truly impressive number of situations [7–54].

Recent progress in this field has allowed for the extraction of new types of bounds: namely, bounds on *ratios* of coefficients. For example, if we assume that the theory in (1.1) is weakly coupled at least up to the scale $M$, then we find [40]

$$-\frac{\sim 6.86}{M^2} \leq \frac{g_3}{g_2} \leq \frac{2}{M^2}\,. \tag{1.2}$$

The appearance of two-sided bounds is a significant advance from the previous linear inequalities – for the first time, it is possible to find closed regions for the space of allowed theories. This brings up the exciting possibility, inspired by the conformal bootstrap, that there may be interesting or important theories lying on the boundary of the allowed region. So far, it is not clear if this is the case. A few such "extremal amplitudes" have been found, but the ones which have been identified as actual UV completions have always arisen from integrating out a single massive particle at tree-level. The other extremal amplitude which has been identified is the "*stu*-pole" [40], which may be problematic because it has an infinite number of states below a given mass (see appendix D of [50] for a discussion). Additionally, the bounds for pion scattering at large $N$ exhibit a kink which is a candidate for large-$N$ QCD, but more study is needed [53, 55]. Indeed, in a number of cases, interesting theories are living inside the allowed region rather than at the boundary – see [49–51, 56, 57]. Given this, we wonder if there is a stronger set of assumptions – perhaps not universal but still satisfied by interesting theories – which shrinks the allowed region such that these theories are on the boundary.

The goal of this paper is to systematically explore some alternate assumptions, and compare the resulting bounds. Our toy model will be a single complex scalar which will be rich enough to explore a wide array of possible situations. Among these are different low-energy theories. In particular, this allows the theory to be coupled to gravity, electromagnetism, both, or neither. These may be thought of as "low-energy assumptions." In practice they determine what poles appear in the low-energy amplitudes. Complex scalars will be more convenient than real scalars for our purposes because of the existence of a $U(1)$ symmetry which can be gauged. A number of interesting Swampland questions are related to the existence of internal symmetries. For example, including gravity but not electromagnetism in the low-energy theory would seem to violate the no-global symmetries conjecture, and making the scalar's electric charge smaller than its mass would seem to violate the weak gravity conjecture. In principle, one might hope to find no allowed regions for these scenarios, though our results are weaker. In the present case we derive several bounds involving the gauge coupling $e^2$, the gravitational coupling $G$, and the coefficients of various contact interactions $g_{a,b}$. We also find a case where a coefficient is only allowed to be negative in the presence of a gauge field, which might be interpreted at saying that its negativity is linked to the absence of a global symmetry.

The other set of assumptions that we will play around with might be thought of as "high-energy assumptions." These include the Regge behavior of the amplitude. It is known that for gapped QFTs, the amplitudes satisfy [58, 59]

$$\lim_{|s|\to\infty} \frac{A(s,t)}{|s|^2} \;=\; 0\,. \tag{1.3}$$

This bound is expected but unproven in a number of other scenarios, which importantly include theories with massless particles. See also [60] for an interesting discussion of a related condition on the growth rate of the amplitude. In the specific case of a massive scalar coupled to gravity, it was proven for $d > 4$ in [61]. That paper made the intriguing observation that in theories of gravity in $d > 4$, certain "smeared" amplitudes should have improved high-energy behavior, namely

$$\lim_{|s|\to\infty} \frac{\int_0^{q_0} dq f(q) A(s,-q^2)}{|s|} \;=\; 0\,. \tag{1.4}$$

where $f(q)$ is required to have certain behavior as $q$ approaches 0 or $q_0$. One of the primary motivations of this paper is to understand how this assumption strengthens the bounds[1]. More generally, we can study how the bounds depend on the Regge behavior. Let us stress that we do not mean to argue that this Regge behavior holds universally – if we derive bounds by assuming a certain Regge behavior, then we think of those bounds as applying to "the class of amplitudes with that Regge behavior," regardless of whether a weaker Regge bound is possible in principle. In this paper, we will primarily be interested in exploring theories whose amplitudes grow slower than $s^2$, $s^1$, and $s^0$ at high-energies– this will lead to the use of doubly-subtracted (2DSR), singly-subtracted (1SDR), and zero-subtracted (0SDR) dispersion relations, respectively.

As we shall see, improving the Regge behavior generally improves the bounds. However, in most cases, we see little or no improvement when going from 2SDR to 1SDR. The reason is rather simple: for singly-subtracted dispersion relations, the left-handed and right-handed branch cuts combine with the opposite sign, in contrast to what happens for doubly-subtracted and zero-subtracted dispersion relations. This complicates the positivity conditions of these dispersion relations, essentially giving useless sum rules for certain dispersive quantities. We find empirically that the extra null constraints derived using 1SDR are also useless in the sense that they do not strengthen the bounds at all[2]. This issue motivates another assumption: that the $++ \to ++$ amplitude has no imaginary part on the positive $s$-axis. This assumption holds in certain weakly-coupled tree-level completions, and it is also exactly analogous to what happens in pion scattering at large-$N$ – in that case, the contribution of a certain isospin structure is subleading in $N$, allowing one to ignore the contribution of the left-handed cut in one of the amplitudes [55].[3] The

---

[1]With singly-subtracted dispersion relations, two-derivative terms become dispersive. But real scalars, photons, and gravitons all have no two-derivative contributions to the $2 \to 2$ amplitude. This is one primary motivation to consider complex scalars.

[2]At least they are useless for the methods available for positivity bounds – it would be interesting to see if they have a role to play in the $S$-matrix bootstrap for instance, where non-linearity unitarity is considered.

[3]Note that pion scattering at large-$N$ is also believed to have $< s^1$ Regge behavior [55] It would be interesting to try to understand if improved Regge behavior is related to the absence of a left-handed cut.

direct result of this assumption is to set the $\rho^{(2)} \equiv \rho_J^{+++}(s)$ spectral density to zero for $s > 0$, which removes the difference of two positive spectral densities. We shall show that with this extra assumption, singly subtracted dispersion relations are stronger than doubly subtracted.

We have provisionally given our assumption the name **$t$-channel dominance**, because the assumption that $\rho^{(2)} \ll \rho^{+-+-}$ and $\rho^{+--+}$ is equivalent to the assumption that the $\mathcal{A}^{++++}$ amplitude is dominated by $t$-channel, rather than $s$-channel exchanges. As we discuss below, this is related to the lack of charge-two states, and also to the requirement that the UV completion be weakly coupled, as loops will inevitably contribute to $\rho^{(2)}$. It is also interesting to note that "$t$-channel dominance" has also appeared in the literature (see section 5.1 of [62]) to describe the related idea that, in systems with different channels, such as pion scattering, the amplitude will be dominated at large-$s$ by the right-most Regge pole. Regge poles are singularities in the high-energy amplitude which appear as if they arose from the exchange of a single particle in the $t$-channel; thus at high-energies the amplitude is characterized by definite quantum numbers in the $t$-channel.

Ultimately, the purpose of this paper is simply to compare the bounds obtained using different assumptions, without consideration for when / whether any particular assumption applies. Our most interesting findings include:

- Going from 2SDR to 1SDR does not substantially improve the bounds, but going from 1SDR to 1SDR + $t$-channel dominance does.

- In the absence of massless exchanges, our bounds for 1SDR + $t$-channel dominance are identical to those explored for pion scattering in [53, 55]. This is simply because both cases can be reduced to a single partially $(s - t$ only$)$ crossing-symmetric function.

- There are coefficients which are positive in the forward limit but allowed to be negative in the presence of gravity [63]. We study one such coefficient, $g_{0,2}$, and find that assuming 1SDR + $t$-channel dominance restores positivity in the absence of a gauge field. We conclude that if $g_{0,2}$ is negative, then the $U(1)$ symmetry of the complex scalar must be gauged.

- Using 1SDR and $t$-channel dominance, we find a bound on the strength of the gravitational coupling in terms of the gauge field and leading dispersive coefficient:

$$G \leq 10.6618e^2 + 0.0367g_{0,1} \,. \tag{1.5}$$

## 2 Comparing Regge behavior for real scalars

Let us review some of these topics in the simpler case of a massless real scalar, which also gives a clean illustration of how fewer subtractions strengthens the bounds. This is exactly the case considered in [40]. We will use conventions where the metric is mostly-plus and momenta are all-ingoing. This leads us to define the Mandelstam invariants

$$s = -(p_1 + p_2)^2 \,, \quad t = -(p_1 + p_3)^2 \,, \quad u = -(p_1 + p_4)^2 \,. \tag{2.1}$$

The amplitude then includes the gravitational interactions, plus an infinite series of contact terms:

$$\mathcal{A}_{\text{low}} = 8\pi G_N \left[ \frac{st}{u} + \frac{su}{t} + \frac{tu}{s} \right] + \sum_{a,b\geq 0} f_{a,b}(s^2 + t^2 + u^2)^a (stu)^b . \tag{2.2}$$

We have assumed that the cubic self-coupling is zero here. We parametrize the unknown high-energy amplitude using partial waves:

$$\mathcal{A}(s,t) = \sum_{J \text{ even}} n_J^{(d)} f_J(s) P_J \left( 1 + \frac{2t}{s} \right) , \tag{2.3}$$

where $P_J(x)$ are the dimension-dependent Gegenbauer polynomials, and $f_J(s)$ is the partial wave density, and the constant $n_J^{(d)}$ is defined by

$$n_J^{(d)} = \frac{(4\pi)^{d/2}(d + 2J - 3)\Gamma(d + J - 3)}{\pi \Gamma(\frac{d-2}{2})\Gamma(J + 1)} . \tag{2.4}$$

We pursue the typical strategy, which is to apply subtracted dispersion relations of the form

$$\oint \frac{ds'}{2\pi i} \frac{\mathcal{A}(s',t)}{s'^{k+1}} = 0 . \tag{2.5}$$

The values of $k$ for which this is valid depend on the high-energy behavior of $\mathcal{A}(s,t)$. In general, if

$$\lim_{|s|\to\infty} \frac{\mathcal{A}(s,t)}{s^k} = 0 \tag{2.6}$$

then (2.5) will hold. We refer to equation (2.5) as a $k$-subtracted dispersion relation, or a $k$SDR. As we decrease the number of subtractions, we find that more coefficients appear in the dispersion relation.

**Sum rules and null constraints** If there is no gravity then there are no poles at $t = 0$ coming from massless exchanges. In this case, we can expand the sum rules in the forward limit to derive dispersion relations for each of the couplings. We denote the sum-integrals using the bracket:

$$\langle X(s,t) \rangle = \sum_{J \text{ even}} n_J^{(d)} \int_{M^2}^{\infty} \frac{ds'}{\pi} \left( \rho_J(s') X(s',t) \right) . \tag{2.7}$$

Then we have the following dispersive definitions of the couplings:

$$
\begin{aligned}
f_{0,0} &= \left\langle \frac{2}{s'} \right\rangle & f_{1,0} &= \left\langle \frac{1}{s'^3} \right\rangle \\
f_{0,1} &= \left\langle \frac{-4\mathcal{J}^2 + 3d - 6}{(d-2)s'^4} \right\rangle & f_{2,0} &= \left\langle \frac{1}{2s'^5} \right\rangle \\
f_{1,1} &= \left\langle \frac{-4\mathcal{J}^2 + 5d - 10}{2(d-2)s'^6} \right\rangle & f_{3,0} &= \left\langle \frac{1}{4s'^7} \right\rangle \\
f_{0,2} &= \left\langle \frac{4\mathcal{J}^4 + (8 - 14d)\mathcal{J}^2 + 3d^2 - 6d}{d(d-2)s'^7} \right\rangle
\end{aligned}
\tag{2.8}
$$

where $\mathcal{J}^2 = J(J + d - 3)$. The $f_{0,0}$ relation is only valid when there are zero-subtracted relations. Some of the coefficients can be simplified using null constraints, but we have written them here in a way which is valid for all numbers of subtractions (assuming they are dispersive at all).

It is possible to write improved null constraints, which are functions of $t$, by subtracting an infinite number of coefficients from the sum rules (see appendix A of [40]).

$$
\begin{aligned}
X_0 &= \left\langle \frac{(2s' + t)P_J\left(1 + \frac{2t}{s'}\right)}{(s' + t)} - \frac{2s'^2}{s'^2 - t^2} \right\rangle \\
X_2 &= \left\langle \frac{(2s' + t)\left(P_J\left(1 + \frac{2t}{s'}\right) - 1\right)}{s't^2(s' + t)^2} - \frac{4\mathcal{J}^2}{(d - 2)s'(s'^2 - t^2)} \right\rangle
\end{aligned}
\tag{2.9}
$$

Similarly, we can obtain expressions for $X_4$, $X_6$, etc. $X_0 = 0$ is only valid for 0 subtractions, $X_2 = 0$ is valid when there are two or fewer subtractions, and so on.

**Bounds** Using these definitions, we can derive bounds and compare for different numbers of subtractions. In figure 1, we do this for two different combinations of coefficients for the real scalar theory defined in equation (2.2). In (a), we consider three coefficients which are familiar from [40] (the coefficients are named $g_2$, $g_3$, and $g_4$ in that work). The kink on the top right corresponds to integrating out a massive scalar at tree-level,

$$
\begin{aligned}
A_{\text{scalar}} &= \frac{1}{m^2 - s} + \frac{1}{m^2 - t} + \frac{1}{m^2 - u} \\
&= \frac{3}{m^2} + \frac{1}{m^6}(s^2 + t^2 + u^2) + \frac{3}{m^8}stu + \cdots
\end{aligned}
\tag{2.10}
$$

While the new kink in the orange plot is the familiar "$stu$"-pole amplitude,

$$
A_{stu-\text{pole}} = \frac{m^4}{(m^2 - s)(m^2 - t)(m^2 - u)} = \frac{1}{m^2} + \frac{1}{2m^6}(s^2 + t^2 + u^2) + \frac{1}{m^8}stu + \cdots
\tag{2.11}
$$

The kink at the top left of the blue plot disappears when we restrict to zero-subtracted dispersion relations. That amplitude is a linear combination of the other two amplitudes, $A_{\text{sub}} = A_{stu} - \gamma A_{\text{scalar}}$, where $\gamma \simeq .462$ is a positive constant. It is interesting to try to understand why this kink is ruled out while the $A_{\text{scalar}}$ kink is not. From one point of view, it is clear. With zero-subtracted dispersion relations, it is possible to derive a bound

$$
\frac{f_{1,0}}{f_{0,0}} \leq \frac{1}{2M^4} .
\tag{2.12}
$$

This bound is saturated by $A_{stu}$, and will be violated by $A_{stu} - \alpha A_{\text{scalar}}$ for any positive $\alpha$.

However, it is curious that both $A_{\text{scalar}}$ and $A_{\text{sub}}$ are both marginal with respect to the zero-subtracted Regge bound: both have terms of order $s^0$ in the large-$s$ limit. Strictly speaking, this means that both are ruled out, but we find that the $A_{\text{scalar}}$ is allowed by the numerics. We believe this means that $A_{\text{scalar}}$ can be "fixed" by modifying the amplitude in a way that does not change the low-lying EFT coefficients, and that respects the Regge

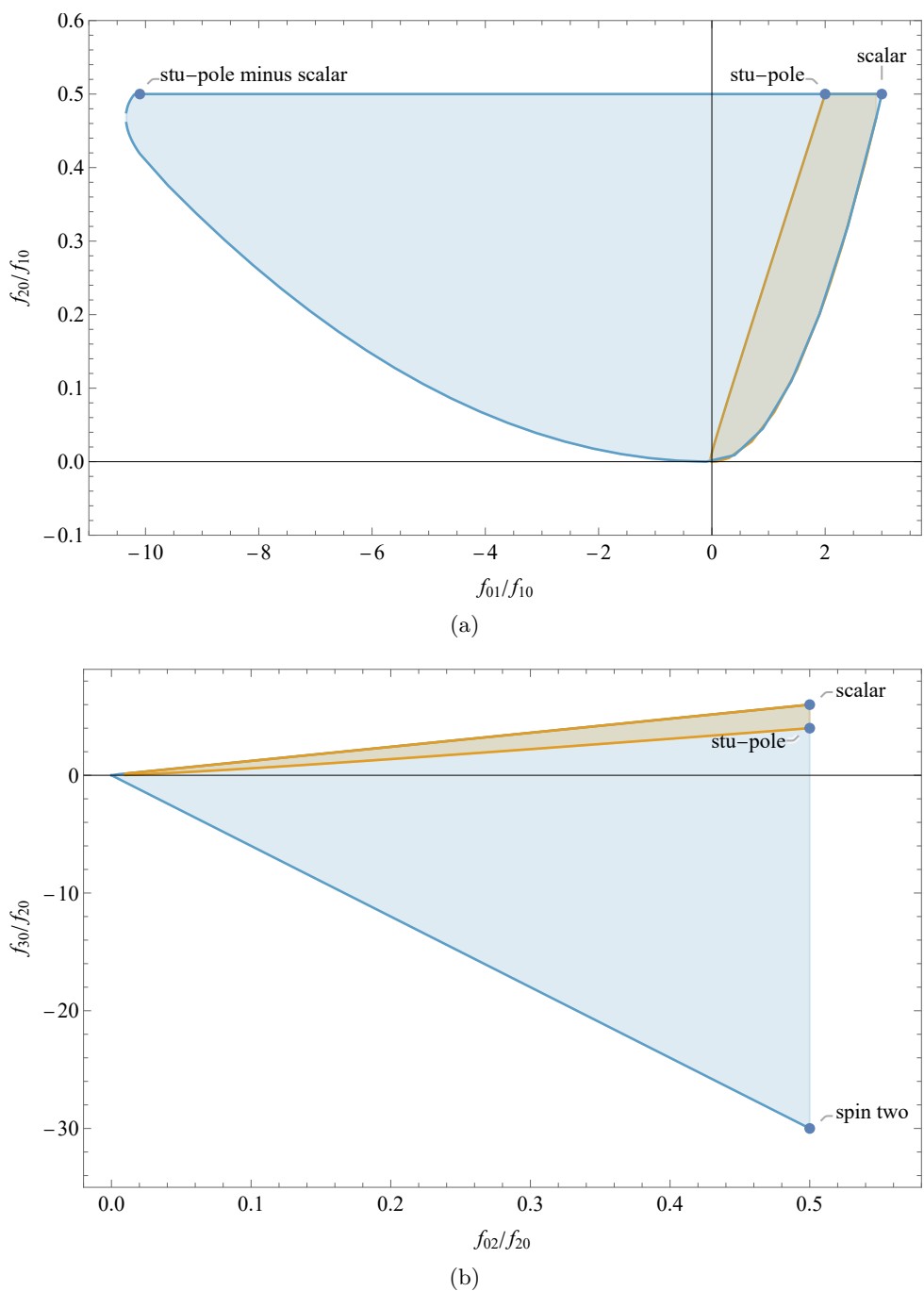

**Figure 1**: Allowed regions for the real scalar theory for $d = 4$. We see that zero-subtracted dispersion relations (orange/smaller) give stronger bounds and a smaller allowed region than doubly-subtracted (blue/larger).

behavior and crossing. Figure 1 (a) seems to be telling us that this is possible for $A_{\rm scalar}$ but not for $A_{\rm sub}$. How this happens in detail would be interesting to explore further.

It is also worth commenting on figure 1 (b), which includes coefficients with dimension 10. The right side of the graph displays three kinks: the top one is $A_{\rm scalar}$ and the second one, which is only a kink using 0SDR, is $A_{stu}$. The bottom kink can be analytically determined as well: it is saturated by the function

$$A_{\rm spin-2} \;=\; \frac{1}{m^4}\left(\frac{s\,m^2 - 6\,t\,u}{m^2 - s} + \frac{t\,m^2 - 6\,s\,u}{m^2 - t} + \frac{u\,m^2 - 6\,s\,t}{m^2 - u}\right). \qquad (2.13)$$

The spectral densities of $A_{\rm spin-2}$ are only non-zero for spin $J = 2$. Up to contact terms, this is the unique amplitude with only a single spectral density and which takes the form of a polynomial times $A_{stu-\rm pole}$ – any other amplitude either changes the spectral densities, or only amounts to contact terms. In fact, this appears to be a general feature: up to the addition of contact terms (which do not affect the spectral density), there is a unique amplitude of the form polynomial $\times A_{stu-\rm pole}$ which is only supported on a single spectral density and which has poles only at $M = m$. It is perhaps not so surprising that such amplitudes would be extremal in the plots, as they are extremal in the sense that every spectral density except one is set to zero– the lower bound. We expect that there is a plot where every such amplitude lies at a kink, and we have checked that this is the case for spin-4 and spin-6, but we do not have a general proof of this.

## 3 Complex scalars

Now we turn to a richer example, which is the theory of a complex scalar field. This is described by the Lagrangian

$$\begin{aligned}
\mathcal{L}_{\rm EFT} \;=\; & -(D_\mu\phi)^\dagger(D^\mu\phi) - m^2\phi^\dagger\phi + \frac{g_{0,0}}{4}(\phi^\dagger\phi)^2 + g_{0,1}(\partial_\mu\phi^\dagger\partial^\mu\phi)(\phi^\dagger\phi) \\
& - g_{1,0}(\partial_\mu\phi^\dagger\partial^\mu\phi)^2 + \left(g_{0,2} + \frac{1}{2}g_{1,0}\right)(\partial_\mu\phi\partial^\mu\phi)(\partial_\mu\phi^\dagger\partial^\mu\phi^\dagger) + \cdots
\end{aligned} \qquad (3.1)$$

The low-energy Lagrangian includes an infinite series of higher-derivative interactions. In this paper, we shall only be interested in the interactions which contribute to the scalar four-point function. In addition to the higher-derivative contact terms, there are also photon exchanges induced by the interactions coming from $D_\mu = \partial_\mu - ieA_\mu$. We allow gravitational exchanges as well by expanding the metric $g_{\mu\nu} = \eta_{\mu\nu} + \kappa h_{\mu\nu}$, with

$$\kappa^2 = 32\pi G. \qquad (3.2)$$

The methods employed in this paper apply only to weakly coupled theories. Specifically, we consider $G$, $e^2$, and all contact terms to be order-$\epsilon$, and we take the amplitude to be the $\epsilon \to 0$ limit of a family of amplitudes. This also removes the need to promote all $\partial_\mu \to D_\mu$, – the result would be terms of order $g_{a,b}e$, which is subleading in $\epsilon$. An analogous statement is true of higher-derivative non-minimal couplings of the graviton to the scalar. Low-energy loops could also introduce order $\epsilon^2$ corrections, which we ignore as well.

There are three possible scalar four-point amplitudes, depending on the charges of the particles:

$$\mathcal{A}_{\text{EFT}}^{++++}(s,t) = \kappa^2\, h(s,t) + e^2\left[\frac{s-u}{t} + \frac{s-t}{u}\right] + \sum_{a,b\geq 0} g_{a,b}(tu)^a(t+u)^b\,,$$

$$\mathcal{A}_{\text{EFT}}^{+--+}(s,t) = \kappa^2\, h(t,s) + e^2\left[\frac{t-u}{s} + \frac{t-s}{u}\right] + \sum_{a,b\geq 0} g_{a,b}(su)^a(s+u)^b \qquad (3.3)$$

$$\mathcal{A}_{\text{EFT}}^{+-+-}(s,t) = \kappa^2\, h(u,t) + e^2\left[\frac{u-s}{t} + \frac{u-t}{s}\right] + \sum_{a,b\geq 0} g_{a,b}(ts)^a(t+s)^b\,.$$

where we have defined the function

$$h(s,t) = -\frac{(t+u)\left(4s^2 + 4t^2 + 4u^2 - (d^2 - 6d + 16)tu\right)}{32tu} + \frac{3m^4(t+u)}{2tu} + \frac{(d-6)m^2}{2}. \quad (3.4)$$

## 3.1 Partial waves

The EFT only describes the theory up to the scale $M$– above that, the amplitude is determined by an unknown UV completion, which we describe with the partial wave expansion:

$$\mathcal{A}(s,t) = \sum_J n_J^{(d)} f_J(s) P_J\left(1 + \frac{2t}{s - 4m^2}\right) \qquad (3.5)$$

where

$$n_J^{(d)} = \frac{(4\pi)^{d/2}(d + 2J - 3)\Gamma(d + J - 3)}{\pi\Gamma\left(\frac{d-2}{2}\right)\Gamma(J + 1)} \qquad (3.6)$$

is a dimension-dependent prefactor and $P_J(x)$ are proportional to Gegenbauer polynomials [64, 65]:

$$P_J(x) = {}_2F_1(-J, J + d - 3, \frac{d-2}{2}; \frac{1-x}{2}) = \frac{\Gamma(1+J)\Gamma(d-3)}{\Gamma(J + d - 3)} C_J^{\left(\frac{d-3}{2}\right)}(x)\,. \qquad (3.7)$$

The result is three partial wave expansions with three different partial wave densities:

$$\mathcal{A}^{++++} = \sum_{J=0,2,4,\ldots} n_J^{(d)}\, f_J^{++++}(s)\, P_J\left(1 + \frac{2t}{s - 4m^2}\right),$$

$$\mathcal{A}^{+--+} = \sum_{J=0,1,2,\ldots} n_J^{(d)}\, f_J^{+--+}(s)\, P_J\left(1 + \frac{2t}{s - 4m^2}\right), \qquad (3.8)$$

$$\mathcal{A}^{+-+-} = \sum_{J=0,1,2,\ldots} n_J^{(d)}\, f_J^{+-+-}(s)\, P_J\left(1 + \frac{2t}{s - 4m^2}\right).$$

Note that the sum in the $++ \to ++$ amplitude only runs over even spins because its ingoing and outgoing states are identical.

### 3.1.1 Unitarity

The dynamical information of these amplitudes is contained in the partial wave densities $f_J(s)$, which have imaginary parts

$$\rho_J(s) \;=\; \frac{(s-4m^2)^{\frac{d-3}{2}}}{\sqrt{s}}\mathrm{Im}f_J(s)\,. \tag{3.9}$$

In the physical region $s > 4m^2$, the spectral densities of the forward amplitudes are required by unitarity to be positive:

$$\rho_J^{++++}(s) \;\geqslant\; 0\,, \qquad\qquad \rho_J^{+-+-}(s) \;\geqslant\; 0. \tag{3.10}$$

Furthermore, crossing symmetry plus the identity $P_J(-x) = (-1)^J P_J(x)$ imply

$$\rho_J^{+--+}(s) \;=\; (-1)^J \rho_J^{+-+-}(s)\,. \tag{3.11}$$

The end result is three positive spectral densities:

$$\rho_J^{(2)}(s) \equiv \rho_J^{++++}(s) \;\geqslant\; 0, \qquad\qquad \text{nonzero for } J \text{ even}, \tag{3.12}$$

$$\rho_J^{(0,+)}(s) \equiv \frac{\rho_J^{+-+-}(s) + \rho_J^{+--+}(s)}{2} \;\geqslant\; 0, \qquad\qquad \text{nonzero for } J \text{ even}, \tag{3.13}$$

$$\rho_J^{(0,-)}(s) \equiv \frac{\rho_J^{+-+-}(s) - \rho_J^{+--+}(s)}{2} \;\geqslant\; 0, \qquad\qquad \text{nonzero for } J \text{ odd}. \tag{3.14}$$

where the superscripts indicate the charge $Q = 0, 2$ and the parity $\pm$ describes whether an even/odd virtual state is exchanged in the $s$-channel.

### 3.2 Dispersion relations

From here on we set the scalar mass $m = 0$. For the massless case, we define the sum rules by

$$C_k(t) \;=\; \oint_\infty \frac{ds'}{2\pi i} \frac{c_1 \mathcal{A}^{++++}(s',t) + c_2 \mathcal{A}^{+--+}(s',t) + c_3 \mathcal{A}^{+-+-}(s',t)}{s'^{k+1}}. \tag{3.15}$$

The assumption that the amplitude is analytic in the $s$-plane allows us to deform the contour to the real axis. We have further assumed that theory is weakly-coupled up to the scale $M^2$, meaning that for $s \lesssim M^2$, the amplitude is described by the tree-level EFT result given in (3.3). The low-energy part of the contour integral therefore is given by the residues of the poles which arise from substituting the amplitudes (3.3) into the dispersion integral:

$$C_k^{\text{low}}(t) = \left(\operatorname*{Res}_{s'=0} + \operatorname*{Res}_{s'=-t}\right)\left[\frac{c_1 \mathcal{A}^{++++}(s',t) + c_2 \mathcal{A}^{+--+}(s',t) + c_3 \mathcal{A}^{+-+-}(s',t)}{s'^{k+1}}\right]. \tag{3.16}$$

For example, we find the low-energy part of the 2SDR to be:

$$\begin{aligned} C_2^{\text{low}}(t) \;=\; & -\frac{8\pi G(c_1 + c_3)}{t} + (c_1 + c_3)g_{0,2} - c_2 g_{1,0} \\ & + t\left(3c_3 g_{0,3} + (c_1 + c_2 + c_3)g_{1,1}\right) + \cdots \end{aligned} \tag{3.17}$$

For the high-energy part, collapsing the contour gives us the discontinuity over the right- and left-handed cuts. The result of combining these using crossing is:

$$C_k^{\text{high}}(t) = \int_{M^2}^{\infty} \frac{ds'}{\pi} \text{Im}\left[ \mathcal{A}^{++++}(s',t) \left( \frac{c_1}{s'^{k+1}} - \frac{c_3}{(-s'-t)^{k+1}} \right) \right.$$

$$+ \mathcal{A}^{+--+}(s',t) \left( \frac{c_2}{s'^{k+1}} - \frac{c_2}{(-s'-t)^{k+1}} \right) \qquad (3.18)$$

$$\left. + \mathcal{A}^{+-+-}(s',t) \left( \frac{c_3}{s'^{k+1}} - \frac{c_1}{(-s'-t)^{k+1}} \right) \right].$$

Using the partial wave expansion, this becomes

$$C_k^{\text{high}}(t) = \left\langle P_J\left(1+\frac{2t}{s'}\right)\left( \frac{c_1}{s'^{k+1}} - \frac{c_3}{(-s'-t)^{k+1}} \right) \right\rangle_2$$

$$+ \left\langle P_J\left(1+\frac{2t}{s'}\right)\left( \frac{c_2+c_3}{s'^{k+1}} - \frac{c_1+c_2}{(-s'-t)^{k+1}} \right) \right\rangle_+ \qquad (3.19)$$

$$+ \left\langle P_J\left(1+\frac{2t}{s'}\right)\left( \frac{c_3-c_2}{s'^{k+1}} - \frac{c_1-c_2}{(-s'-t)^{k+1}} \right) \right\rangle_-$$

where we define the averages

$$\langle X(s,t) \rangle_2 = \int_{M^2}^{\infty} \frac{ds'}{\pi} \sum_{J=0,2,4,\dots} s'^{\frac{4-d}{2}} n_J^{(d)} \rho_J^{(2)}(s') X(s',t), \qquad (3.20)$$

$$\langle X(s,t) \rangle_+ = \int_{M^2}^{\infty} \frac{ds'}{\pi} \sum_{J=0,1,2,\dots} s'^{\frac{4-d}{2}} n_J^{(d)} \rho_J^{(0,+)}(s') X(s',t), \qquad (3.21)$$

$$\langle X(s,t) \rangle_- = \int_{M^2}^{\infty} \frac{ds'}{\pi} \sum_{J=0,1,2,\dots} s'^{\frac{4-d}{2}} n_J^{(d)} \rho_J^{(0,-)}(s') X(s',t). \qquad (3.22)$$

**Regge-boundedness**  If the amplitude grows slower than $s^k$, then we can use $k$-subtracted dispersion relations (or $k$SDR, for short). That is, if

$$\lim_{|s|\to\infty} \frac{\mathcal{A}(s,t)}{s^k} = 0, \qquad t < 0 \qquad (3.23)$$

then

$$C_k^{\text{low}}(t) = C_k^{\text{high}}(t) \qquad (3.24)$$

In what follows, we will also discuss "smeared bounds". These arise when there exists a function $f(p)$, with $t = -p^2$, such that

$$\lim_{|s|\to\infty} \frac{\int f(p)\mathcal{A}(s,-p^2)dp}{s^k} = 0, \qquad (3.25)$$

in which case we have

$$\int f(p) C_k^{\text{low}}(-p^2)dp = \int f(p) C_k^{\text{high}}(-p^2)dp. \qquad (3.26)$$

Typically this requires that we specify the behavior of $f(p)$ near the limits of integration to guarantee (3.25).

## 4 Bounds from forward limit

We can apply these dispersion relations with differing numbers of subtractions, and we can turn off $G$, $e$, or both, so there are a number of scenarios to consider[4]. We shall divide these by first considering the case with no graviton or photon exchange: in this case, $G = e^2 = 0$ and the forward limit sum rules apply and can be used to bound the contact interactions $g_{a,b}$. The case with massless exchanges, where we need to consider smeared dispersion relations, will be considered in section 5.

### 4.1 Sum rules and null constraints

With $G = e^2 = 0$, the poles in the amplitude from massless exchanges will disappear. We can derive all sum rules and null constraints by expanding the dispersion relations in the forward limit $t \to 0$. The exact form of the sum rules, as well as the coefficients that appear in them at all, depends on the number of subtractions. Let us comment on three cases: 2SDR, 1SDR, and 0SDR, and compare the bounds obtained in each case:

**2SDR** The standard assumption is that the amplitude grows slower than $s^2$ at large $s$, which implies the validity of doubly-subtracted dispersion relations. In this case, the leading sum rules are:

$$g_{0,2} = \left\langle \frac{1}{s'^3} \right\rangle_2 + \left\langle \frac{1}{s'^3} \right\rangle_+ + \left\langle \frac{1}{s'^3} \right\rangle_- , \qquad g_{1,0} = -\left\langle \frac{2}{s'^3} \right\rangle_+ + \left\langle \frac{2}{s'^3} \right\rangle_- ,$$

$$g_{0,3} = \left\langle -\frac{1}{s'^4} \right\rangle_2 + \left\langle \frac{1}{s'^4} \right\rangle_+ + \left\langle \frac{1}{s'^4} \right\rangle_- , \qquad (4.1)$$

$$g_{1,1} = -\left\langle \frac{4\mathcal{J}^2 - 3d + 6}{(d-2)s'^4} \right\rangle_+ + \left\langle \frac{4\mathcal{J}^2 - 3d + 6}{(d-2)s'^4} \right\rangle_- ,$$

where we have defined $\mathcal{J}^2 = J(J + d - 3)$. All the contact coefficients can be written with an analogous expression, except for $g_{0,0}$ and $g_{0,1}$, which cannot be bounded in this case. We also find null constraints, the first (*i.e.* the one with the lowest power of $1/s'$) of which is

$$0 = \left\langle \frac{2\mathcal{J}^2}{s'^4} \right\rangle_2 + \left\langle \frac{-2\mathcal{J}^2}{s'^4} \right\rangle_+ + \left\langle \frac{6\mathcal{J}^2 - 6d + 12}{s'^4} \right\rangle_- . \qquad (4.2)$$

In practice, including more null constraints will improve the bounds, and we can easily generate a large number.

**1SDR** If we instead make the stronger assumption that the amplitude grows slower than $s$ at large $s$, then we can apply 1SDRs (there is no justification for this assumption, however we can think of this as focusing only on the subset of amplitudes – if any exist – which have this improved behavior). Then we find a sum rule for $g_{0,1}$

$$g_{0,1} = -\left\langle \frac{1}{s'^2} \right\rangle_2 + \left\langle \frac{1}{s'^2} \right\rangle_+ + \left\langle \frac{1}{s'^2} \right\rangle_- . \qquad (4.3)$$

---

[4]Some cases do not need to be considered separately– for instance, $e$ does not appear in any 2SDR so it does not need to be considered separately in that case.

Note that this coefficient is a *difference* of positive brackets, so it is not necessarily positive. This is in contrast to, for instance, $g_{0,2}$ above which is a sum of positive terms. The coefficient $g_{0,0}$ is still unbounded. The other effect of allowing fewer subtractions is that we get more null constraints, including the lowest order ones:

$$0 \;=\; \left\langle \frac{\mathcal{J}^2}{s'^3} \right\rangle_2 + \left\langle \frac{-\mathcal{J}^2}{s'^3} \right\rangle_+ + \left\langle \frac{-\mathcal{J}^2 + 2d - 4}{s'^3} \right\rangle_- , \tag{4.4}$$

$$0 \;=\; \left\langle \frac{\mathcal{J}^2(\mathcal{J}^2 - 2d + 2)}{s'^4} \right\rangle_2 + \left\langle \frac{\mathcal{J}^2(-\mathcal{J}^2 + 2d - 2)}{(d-2)s'^4} \right\rangle_+ + \left\langle \frac{\mathcal{J}^2(-\mathcal{J}^2 + 2d - 2)}{(d-2)s'^4} \right\rangle_- . \tag{4.5}$$

Notice that the sum rule for $g_{0,1}$ has a dominant behavior at large $s'$, compared to the null constraints. When demanding positivity of a functional on a set of constraints, the brackets in (4.3) will give opposite constraints, resulting in a vanishing entry of the functional. Hence without further assumptions $g_{0,1}$ cannot be bounded. A similar argument applies to the first null constraint in (4.4) in the limit of large $s'$ and large $J$.

**0SDR**  Finally if the amplitude grows slower than a constant at large $s$, then we can use 0SDRs. These allow us to define

$$g_{0,0} \;=\; \left\langle \frac{1}{s'} \right\rangle_2 + \left\langle \frac{1}{s'} \right\rangle_+ + \left\langle \frac{1}{s'} \right\rangle_- . \tag{4.6}$$

and null constraints

$$0 \;=\; \left\langle \frac{1}{s'} \right\rangle_2 + \left\langle \frac{-1}{s'} \right\rangle_+ + \left\langle \frac{3}{s'} \right\rangle_- ,$$
$$0 \;=\; \left\langle \frac{2\mathcal{J}^2}{s'^2} \right\rangle_2 + \left\langle \frac{2\mathcal{J}^2 - d + 2}{s'^2} \right\rangle_+ + \left\langle \frac{2\mathcal{J}^2 - d + 2}{s'^2} \right\rangle_- , \tag{4.7}$$
$$0 \;=\; \left\langle \frac{d - 2}{s'^2} \right\rangle_2 + \left\langle \frac{4\mathcal{J}^2 - 2d + 4}{s'^2} \right\rangle_+ + \left\langle \frac{-4\mathcal{J}^2}{s'^2} \right\rangle_- ,$$

plus a number of null constraints appearing at orders $s'^{-3}$ and beyond. It is perhaps interesting to note that in the second of these constraints, every term is positive except for the $\langle \cdots \rangle_\pm$ term at $J = 0$. That means that those terms must be non-zero, as they are responsible for balancing every other term in the sum.

In [53] it was suggested that the amplitude for pion scattering would allow for certain 0SDRs, called ST dispersion relations in that work, but they could not be exploited numerically because they dominate at large $s'$ but oscillate in spin. Such constraints correspond to the last line of (4.7). In order to make use of them, it was shown in [53] that additional null constraints – from the so-called SU disperion relations – are necessary. However this leads to a problem – for pion scattering, 0SDR cannot be universally valid because the pion is a Goldstone boson and hence derivatively coupled. This implies $g_{0,0} = 0$, but equation (4.6) would require that $g_{0,0} > 0$. Thus ST and SU dispersion relations can't both be used. In our case we don't require that the scalar is a goldstone boson, in which case we are safe to assume 0SDR. We discuss the impact of these null constraints in section 5.2.

## 4.2 Results

In figure 2, we have plotted the bounds that follow from the sum rules above. In figure 3 instead we show that allowed regions are shaped by regions where only a subset of spins are allowed to contribute to the partial wave decomposition: only $J = 0$, only $J = 1$ or only $J \geq 2$. Notice that when using $k$SDR, there are no allowed regions with only spins $J = \ell$, with $\ell > k$, as expected since such amplitude would violate the Regge behavior at large $s$.

### 4.2.1 Analytically ruling in

Let us now comment on some of the UV completions that we are able to identify on these plots. We will define

$$\vec{g} = \{g_{0,0}, g_{0,1}, g_{0,2}, g_{1,0}, g_{0,3}, g_{1,1}, g_{0,4}, g_{1,2}, g_{2,0}\} \tag{4.8}$$

**Massive neutral scalar** A simple completion arises from integrating out a massive scalar at tree level, which can couple through the Lagrangian

$$\mathcal{L} = -\frac{1}{2}(\partial_\mu \phi)^\dagger (\partial^\mu \phi) - \frac{1}{2}\partial_\mu \Phi \partial^\mu \Phi - \frac{1}{2}M^2 \Phi^2 + g\phi^\dagger \phi \Phi. \tag{4.9}$$

The amplitude which arises from this is

$$\mathcal{A}_{\text{scalar}}^{++++}(s,t) = \frac{g^2}{M^2 - t} + \frac{g^2}{M^2 - u}. \tag{4.10}$$

We can set $M \to 1$ since it is the only energy scale, and $g \to 1$ since it will not affect the ratios of coefficients. Then we find

$$\vec{g}_{\text{NS}} = \{2, 1, 1, -2, 1, -3, 1, -4, 2\}. \tag{4.11}$$

**Massive charged scalar** Another option is that the massive scalar has charge 2, so that the two incoming scalars can combine into it. This completion provides a counterexample to our "$t$-channel dominance" assumption, so we shall see that it lies outside the allowed region when we make this assumption. The amplitude is

$$\mathcal{A}_{s-\text{pole}}^{++++}(s,t) = \frac{g^2}{M^2 - s}. \tag{4.12}$$

Setting $M \to 1$, we find the coefficients

$$\vec{g}_{\text{CS}} = \{1, -1, 1, 0, -1, 0, 1, 0, 0\}. \tag{4.13}$$

**Massive neutral vector** The charged scalar can also couple to a massive vector through an $A_\mu J^\mu$ term, where $J^\mu = ig(\phi\partial^\mu \phi^\dagger - \phi^\dagger \partial^\mu \phi)$. This gives the amplitude

$$\mathcal{A}_{\text{vector}}^{++++}(s,t) = g^2 \left(\frac{u - s}{M^2 - t} + \frac{t - s}{M^2 - u}\right) \tag{4.14}$$

which leads to

$$\vec{g}_V = \{0, 3, 1, 2, 1, -1, 1, -2, -2\}. \tag{4.15}$$

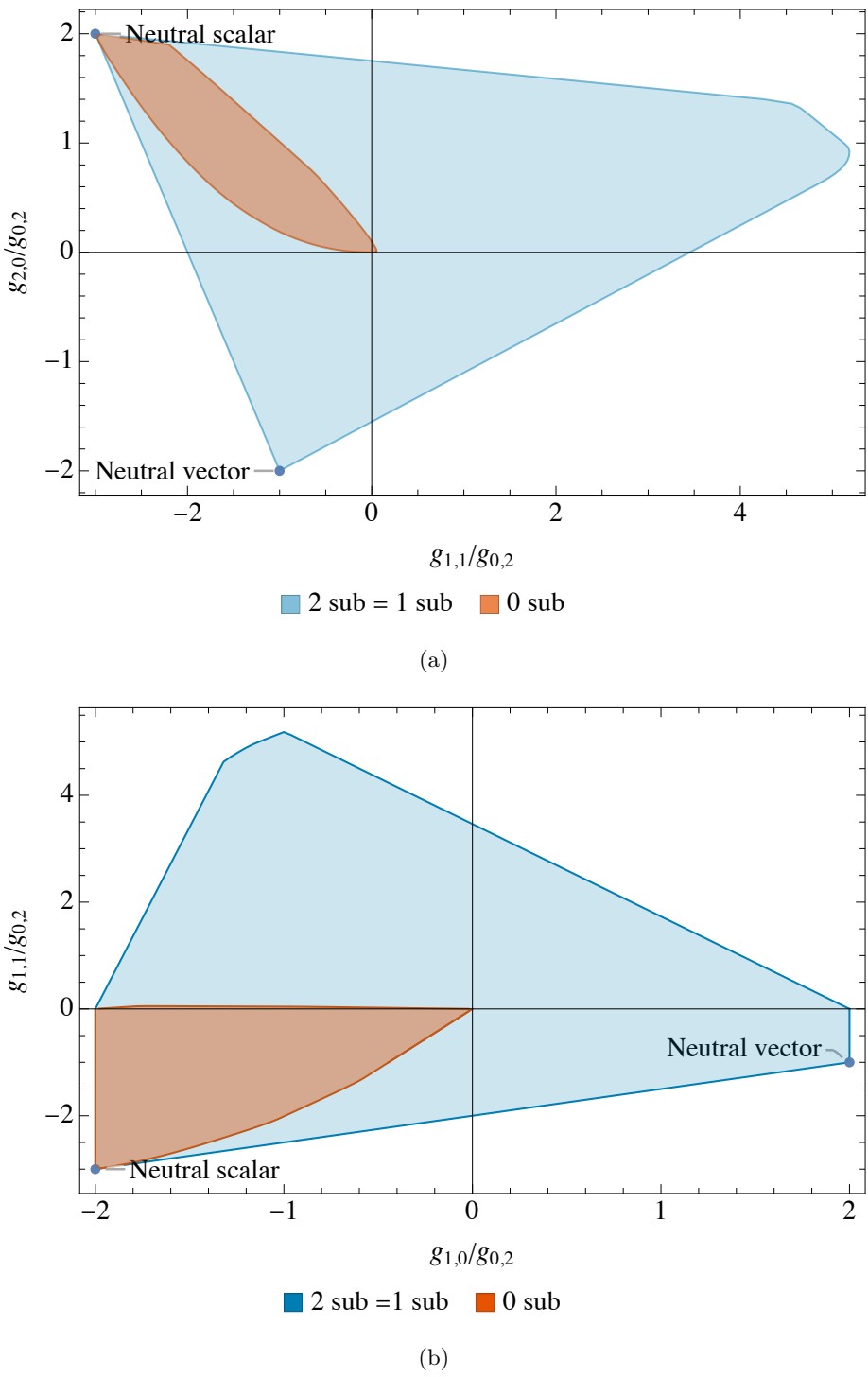

**Figure 2**: Allowed regions from forward-limit bounds for $d = 4$. We can see that the bounds obtained for one and two subtractions are almost identical. The bounds obtained from zero-subtracted dispersion relations are much stronger, however.

**Massive charged vector** Similarly, it is possible to have a charged vector which couples through a term like $A_\mu \phi \partial^\mu \phi$. This leads to the amplitude

$$\mathcal{A}_{\text{c-vector}}^{++++}(s,t) \;=\; g^2 \frac{s}{M^2 - s}\,, \tag{4.16}$$

and leads to

$$\vec{g}_{\text{CV}} \;=\; \{0, -1, 1, 0, -1, 0, 1, 0, 0\}\,. \tag{4.17}$$

This amplitude does not really represent a massive charged vector. In fact it is related to the massive charged scalar by an overall derivative, and it has no spectral density in the spin-1 channel. This may be thought of as a version of the Landau-Yang theorem for charged scalars. Nonetheless it is still an amplitude satisfying our assumptions, so we include it here.

**Massive gravitons** We can naively take the amplitude in equation (2.13), which only has non-zero spectral density for spin-two, and consider the $t - u$ symmetric version:

$$\mathcal{A}_{\text{graviton}} \;=\; g^2 \left( \frac{t^2 - 6\,s\,u}{M^2 - t} + \frac{u^2 - 6\,s\,t}{M^2 - u} \right)\,, \tag{4.18}$$

leading to

$$\vec{g}_{\text{NG}} \;=\; \{0, 0, 7, -2, 1, 9, 1, 2, 2\}\,. \tag{4.19}$$

If we consider a charge-2 graviton with the amplitude

$$\mathcal{A}_{\text{charged graviton}} \;=\; g^2 \frac{s^2 - 6\,t\,u}{M^2 - s}\,, \tag{4.20}$$

we have

$$\vec{g}_{\text{CG}} \;=\; \{0, 0, 1, -6, -1, 6, 1, -6, 0\}\,. \tag{4.21}$$

*tu*-**pole** In [40], it was pointed out that the amplitude

$$\mathcal{A}_{stu} \;=\; \frac{1}{(M^2 - s)(M^2 - t)(M^2 - u)} \tag{4.22}$$

is a crossing symmetric function which satisfies the required Regge behavior. What is more surprising is that this "amplitude" also has positive partial wave densities, leading to the question of whether it is the amplitude of a real theory. This is also $s - u$ symmetric, so it gives an amplitude with coefficients

$$\vec{g}_{stu-\text{pole}} = \{1, 0, 1, -1, 0, -1, 1, -2, 1\}\,. \tag{4.23}$$

There is another amplitude appears at the corners of some of our plots:

$$\mathcal{A}_{tu-\text{pole}}^{++++}(s,t) \;=\; \frac{1}{(M^2 - u)(M^2 - t)}\,. \tag{4.24}$$

This gives coefficients

$$\vec{g}_{tu-\text{pole}} = \{1, 1, 1, -1, 1, -2, 1, -3, 1\}\,. \tag{4.25}$$

***tu*-pole minus scalar***  The $t - u$ pole has all positive spectral densities, so it is possible to subtract the scalar amplitude

$$\mathcal{A}_\alpha(s,t) \;=\; \mathcal{A}_{tu-\text{pole}}(s,t) - \alpha \mathcal{A}_{\text{scalar}}(s,t)\,. \tag{4.26}$$

The maximum value of $\alpha$ occurs where the spin-zero spectral density is zero. In $d = 4$, this was determined by [55] to be $\alpha = \log 2$. Choosing this value, we find

$$\vec{g}_{\text{sub}} \;=\; \{-0.386, 0.307, 0.307, 0.386, 0.307, 0.0794, 0.307, -0.227, -0.386\}\,. \tag{4.27}$$

### 4.2.2   Bounds and $t$-channel dominance

One of the most striking features of the plots given in section 4.2 is that the inclusion of singly-subtracted dispersion relations does not improve the bounds at all. This appears to be a rather general property stemming from the fact that for all of the 1SDR sum rules and null constraints, the terms in $\langle \cdots \rangle_2$ bracket and the $\langle \cdots \rangle_\pm$ bracket have the opposite sign at large $J$. For example, recall that

$$g_{0,1} \;=\; -\left\langle \frac{1}{s'^2} \right\rangle_2 + \left\langle \frac{1}{s'^2} \right\rangle_+ + \left\langle \frac{1}{s'^2} \right\rangle_-\,. \tag{4.28}$$

This is a difference of two positive averages, so it cannot be bounded without further assumptions. One simple assumption which allows us to bound it is simply to assume that

$$\textbf{\textit{t}-channel dominance}: \qquad \rho^{(2)}(s) = 0 \qquad \text{for } s > M^2\,. \tag{4.29}$$

In this case, under this assumption $g_{0,1}$ is positive.

The assumption (4.29) is essentially that the $++ \to ++$ amplitude gets no contribution from $s$-channel exchanges. As such, we shall call our assumption "$t$-channel dominance". It is directly analogous to the assumption, made in [55], that the isospin-two channel does not contribute to pion scattering. In that case, the isospin-two channel is $1/N$ suppressed, so it only holds exactly at large-$N$.

It is natural to ask when $t$-channel dominance actually holds for scalar scattering. If we look at the tree-level completions presented in the last subsection, it is clear that the assumption holds for the massive neutral scalar, neutral vector, and neutral spin-two. However, it is violated by the massive charged scalar, charged vector, and charged spin-two, all of which have an $s$-channel pole. So one consequence of our assumption would be that the coupling of our charged scalar to a hypothetical massive charge-two state is suppressed relative to the coupling of the scalar to massive neutral states. A non-zero value for $\rho^{(2)}$ can also be generated by integrating out neutral matter at loop level. Then $t$-channel dominance will only hold when $\rho^{(2)}$ is small compared to $\rho^{(0,\pm)}$, which will be true if the UV completion is also weakly coupled. These considerations are illustrated in figure 4, where (a) and (b) must be suppressed relative to (c).

Having weakly-coupled charge-two states and a weakly coupled UV completion are relatively simple criteria. The first criterion, the dominance of neutral exchanges over charge-two exchanges is somewhat reminiscent of the weak gravity conjecture [66] and the

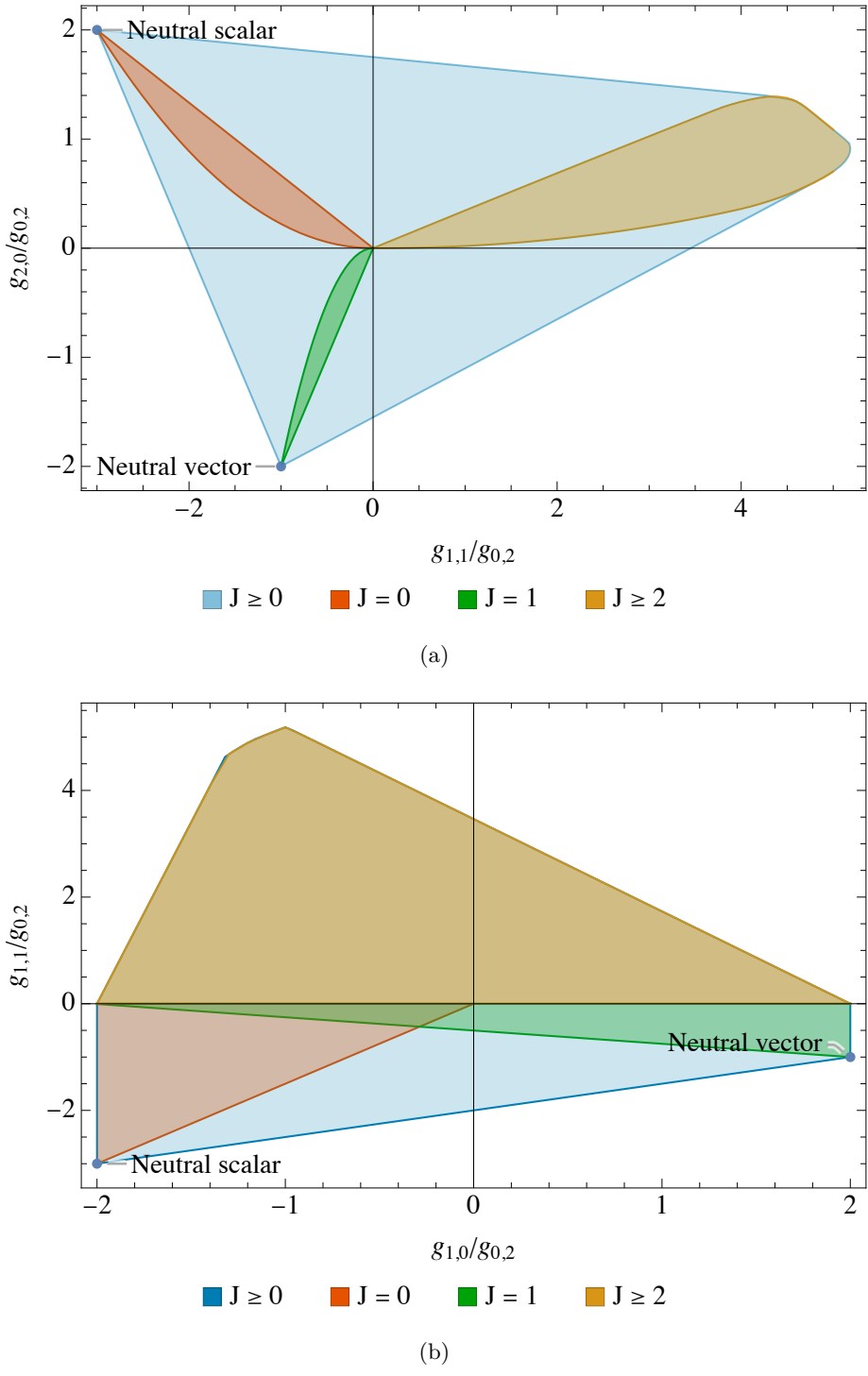

**Figure 3**: Allowed regions from forward-limit bounds for $d = 4$ (assuming 2SDR) requiring positivity only on a subset of spins. The full allowed region is the convex hull of solutions obtained using only scalars, only spin-1 or $J \geq 2$.

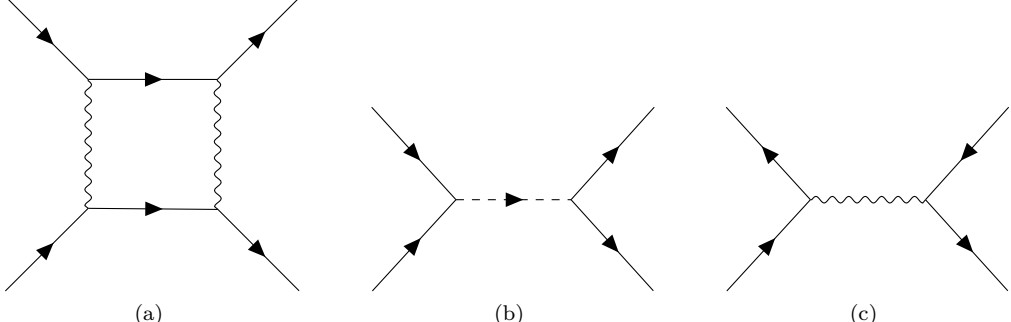

**Figure 4**: Contributions to $\rho^{(2)}$ can include (a) exchange of neutral states through loops or (b) exchange of a charge-two state. The $t$-channel dominance assumption is that these are suppressed relative to $\rho^{(0,\pm)}$, which can be generated by neutral states at tree level (c).

related repulsive force conjecture [67]. If our complex scalar has a mass greater than its charge, it will naturally form charge-two bound states that might contribute to $\rho^{(2)}$. Note that in our analysis we are considering the scalar to be massless, but this can be thought of as an approximation: introducing a small scalar mass $m$ should introduce changes of order $m/M$.

The second assumption, that the UV completion is weakly coupled, is not an assumption in the EFT bootstrap setup, but it appears in a number of examples that extremal EFTs tend to arise from tree-level completions (for instance, in [49] it was observed that loop-level completions typically lie deep inside the allowed regions).

Figures 2, 3, 5, and 6 show the bounds derived with and without the assumption $t$-channel dominance. In particular, figure 6 considers the same coefficients as figure 1 of [55], and can therefore be compared to that work. In figure 6b, we see that the kink observed in [55] is only a corner with the 1SDR plus $t$-channel dominance, which were the exact assumptions used in that paper. We also observe that 0SDRs represent a very strong constraint– in fact, they are too strong for the unknown kink to survive.[5] It would be interesting to systematically include the 0SDR null constraints one at the time to determine if there is a subset of null equations that still gives bounds consistent with the large-$N$ QCD expectations.

## 5  Bounds with massless exchanges

Allowing for massless exchanges introduces poles into the amplitude, making it impossible to expand around the forward limit. This problem has been discussed in [27, 68], but a new perspective was given in [63], where it was shown that bounds can be obtained, even in the presence of the graviton-exchange pole, by integrating the sum rules against a

---

[5]In particular, comparing with figure 6a we see that 0SDRs demand the presence of $J = 0$ states, while the kink was given by solutions without scalars.

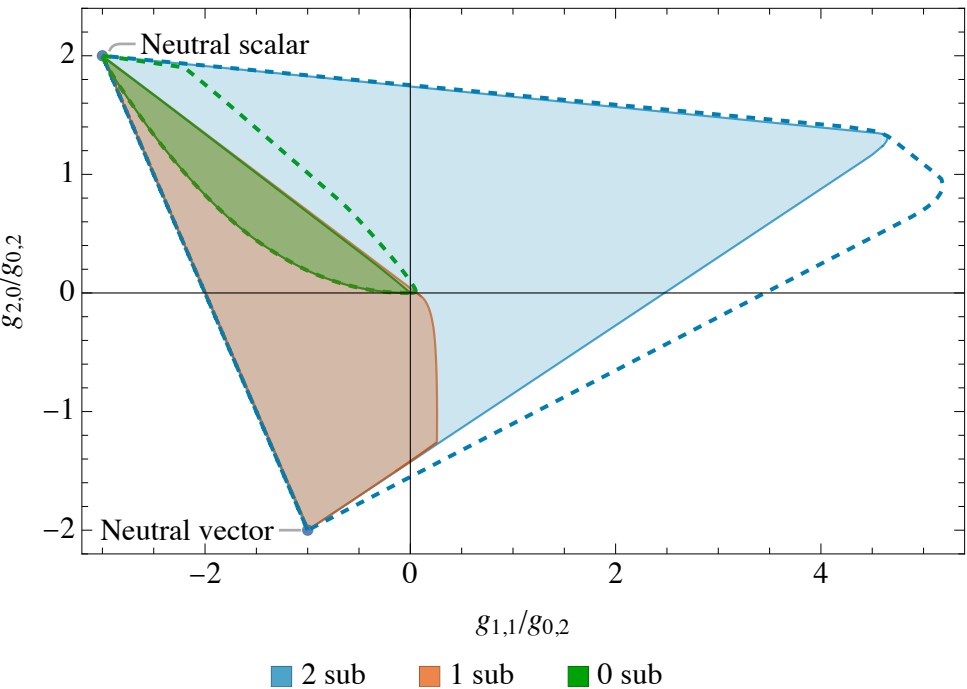

**Figure 5**: Here we have plotted the same coefficients as figure 2a including the assumption of $t$-channel dominance, for 2, 1, and 0 subtractions. The original bounds of figure 2a are included in dashed lines for comparison.

kernel clustered around small but non-zero $t$. These "smeared bounds" require first that we systematically subtract off higher-derivative operators to derive improved sum rules.

## 5.1 Improved sum rules

As it stands, the low-energy part of the sum rule (3.16) contains infinitely many Wilson coefficients. The simplest way to bound a particular ratio is to expand the LHS and RHS around $t = 0$, matching the sides order by order. In the presence of a graviton pole, this does not work because the LHS has a $1/t$ pole but the RHS does not; the presence of the pole ruins the commutativity of the sum-integral and the $t \to 0$ limit. Another method is needed to write a sum rule with a finite number of coefficients – one must systematically subtract almost all coefficients and obtain an improved sum rule which contains finitely many Wilson coefficients [63], using higher-subtracted sum rules which *do* converge in the forward limit. We shall see this below.

### 2SDRs

By taking the following combination, one obtains in the low-energy a simple expression containing only finitely many Wilson coefficients. Recall how the partial wave expansion allows us to write the high-energy part as an infinite sum-integral over all possible values

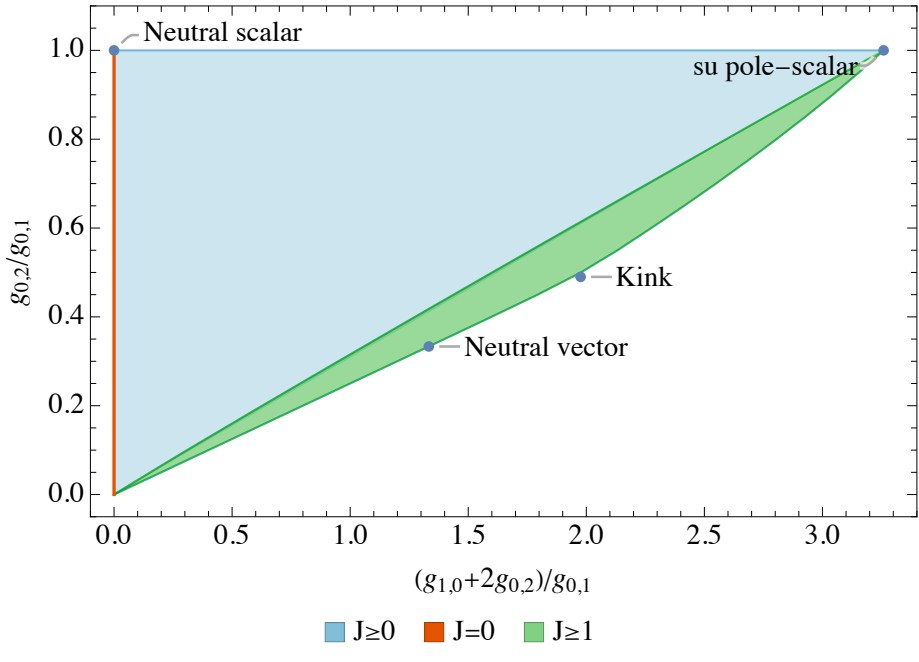

(a) allowed region with 1SDR + $t$-channel dominance

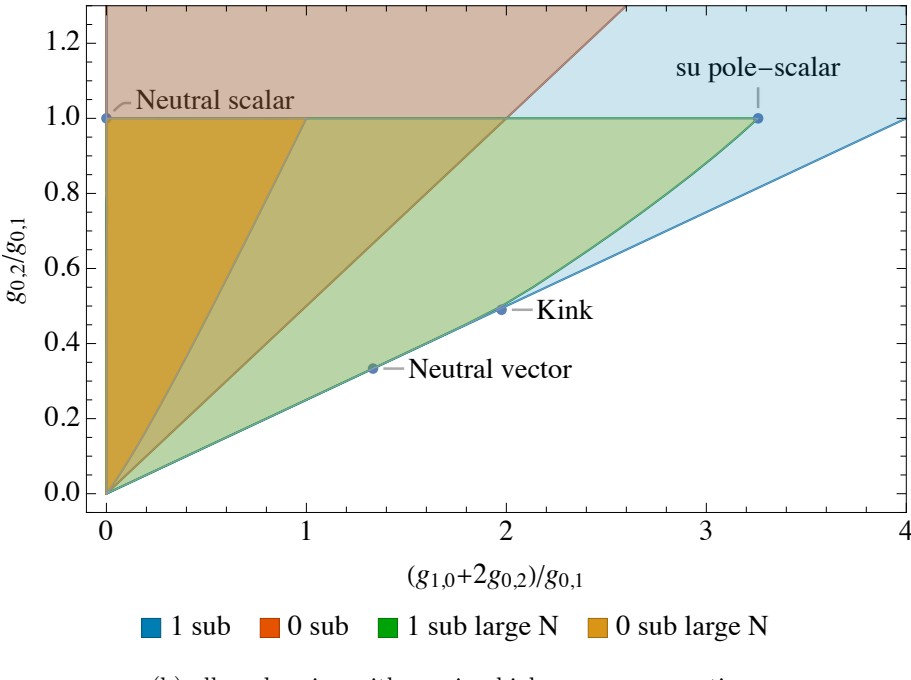

(b) allowed region with varying high-energy assumptions

**Figure 6**: Bounds on the same coefficients as in figure 1 of [55]. We see that it is necessary to impose $t$-channel dominance to see the unknown kink.

of $J$ and $s'$,

$$C_k^{\text{high}}(t) = \left\langle P_J\left(1 + \frac{2t}{s'}\right)\left(\frac{c_1}{s'^{k+1}} - \frac{c_3}{(-s'-t)^{k+1}}\right)\right\rangle_2$$
$$+ \left\langle P_J\left(1 + \frac{2t}{s'}\right)\left(\frac{c_2 + c_3}{s'^{k+1}} - \frac{c_1 + c_2}{(-s'-t)^{k+1}}\right)\right\rangle_+ \qquad (5.1)$$
$$+ \left\langle P_J\left(1 + \frac{2t}{s'}\right)\left(\frac{c_3 - c_2}{s'^{k+1}} - \frac{c_1 - c_2}{(-s'-t)^{k+1}}\right)\right\rangle_- .$$

Then we can ensure that the LHS will include only a *finite* number of EFT coefficients by constructing various *improved sum rules*, such as

$$C_{k,(c_1,c_2,c_3)}^{\text{improved}}(t) \equiv C_{k,(c_1,c_2,c_3)}^{\text{high}}(t) - \frac{1}{2}\sum_{k=3}^{\infty} t^k \left(\frac{\partial^2}{\partial t^2} C_{k,(c_2,c_1,c_3)}^{\text{high}}(t)\bigg|_{t=0}\right), \qquad (5.2)$$

where one should note the different order of coefficients $c_i$ appearing in the sum. This removes higher-order coefficients using an infinite number of forward-limit sum rules, which start at $k > 2$. By subtracting a few more terms we obtain the simple expression

$$-\frac{8\pi(c_1+c_3)G}{t} + (c_1+c_3)g_{0,2} - c_2\,g_{1,0} + t(c_1+c_2+c_3)g_{1,1} =$$
$$= C_{2,(c_1,c_2,c_3)}^{\text{improved}}(t) - 3t\,C_{3,(0,0,c_3)}^{\text{high}}(0) - t^2\frac{\partial}{\partial t}C_{3,(c_1+c_2-\frac{c_3}{2},\frac{c_1+c_2+c_3}{2},\frac{3}{2}c_3)}^{\text{high}}(0). \qquad (5.3)$$

**1SDRs**

Similarly to the previous subsection, we can obtain the following improved single subtracted dispersion relations

$$\frac{2(c_1-c_3)e^2}{t} + \pi G\left(c_3(d-6)d - c_1((d-6)d+16)\right) + (c_3-c_1)\,g_{0,1}$$
$$+ 2c_3tg_{0,2} - (c_1+c_2-c_3)\,tg_{1,0} + (c_1+c_2+c_3)\,t^2g_{1,1} =$$
$$= C_{1,(c_1,c_2,c_3)}^{\text{high}}(t) - \sum_{k=3}^{\infty} t^k\left(\frac{\partial}{\partial t}C_{k,(c_2,c_1,c_3)}^{\text{high}}(t)\bigg|_{t=0}\right) - 3t^2 C_{3,(0,0,c_3)}^{\text{high}}(0). \qquad (5.4)$$

**0SDRs**

Finally, the case of zero-subtracted dispersion relation assumes the simple form

$$(c_1+c_2+c_3)\,g_{0,0} - \pi Gt\left(c_2((d-6)d+24) - c_3(d-6)d+8c_1\right) + 2(c_2-c_3)e^2$$
$$+ (c_2+c_3)\,t^2g_{0,2} - c_1t^2g_{1,0} + (c_3-c_2)\,tg_{0,1} =$$
$$= C_{0,(c_1,c_2,c_3)}^{\text{high}}(t) - \sum_{k=3}^{\infty} t^k\left(C_{k,(c_2,c_1,c_3)}^{\text{high}}(t)\bigg|_{t=0}\right). \qquad (5.5)$$

For each of the above expressions one can find three independent combinations, spanned by independent choices of $c_1$, $c_2$, and $c_3$. In the rest of the section we will refer to these equations by $E_i$, which are defined below in table 1. Note that $E_1$ is a pure null constraint.

| symbol | eq. | $c_1$ | $c_2$ | $c_3$ |
|:---:|:---:|:---:|:---:|:---:|
| $E_1$ | (5.3) | 1 | 0 | $-1$ |
| $E_2$ | (5.3) | 0 | 1 | 0 |
| $E_3$ | (5.3) | 0 | 0 | 1 |
| $E_4$ | (5.4) | 1 | 0 | 1 |
| $E_5$ | (5.4) | 0 | 1 | 0 |
| $E_6$ | (5.4) | 0 | 0 | 1 |
| $E_7$ | (5.5) | 1 | 0 | $-1$ |
| $E_8$ | (5.5) | 0 | 1 | 0 |
| $E_9$ | (5.5) | 0 | 0 | 1 |

**Table 1**: Choice of parameters $c_i$ in the 2SDR, 1SDR and 0SDR.

## 5.2 Smeared functionals

In the rest of the section we will use the convention $t = -p^2$. We obtain bounds on the coefficients $G$, $e$ and $g_{a,b}$ by applying integral functionals to equations (5.3), (5.4) and (5.5). Define $\vec{v}$ to be a nine-component vector whose components are functions of $p$. Then we introduce the functional $\Lambda_I[\vec{v}]$ defined as

$$\Lambda_I[\vec{v}] = \sum_{i \in I} \int_0^1 \phi_i(p) v_i(p) dp \,, \tag{5.6}$$

where we shall use as ansatz for our functional polynomials

$$\phi_i(p) = \sum_{n \in S_i} a_{i,n} p^n (1-p) \,. \tag{5.7}$$

The exponents $S_i$ depend on the equation $E_i$ and size of the numerical problem. Our choices are summarized in appendix A.

If we manage to find a functional $\Lambda_I$ that is non-negative on the RHS of the dispersion relations (5.3), (5.4) and (5.5) for every possible value of $J$ and $s' = m^2$, then we can obtain bounds in the form of inequalities among the low energy parameters in the LHS. The existence of coefficients $a_{i,n}$ that realize this condition is checked numerically by translating the search in a semi-definite programming problem and solving it with SDPB [69]. We look for functionals that are individually positive on the integrand in the RHS with fixed $J$ and $s' \geq M^2$. This includes the asymptotic region

$$J \to \infty, \, s' \to \infty, \quad \text{with fixed} \quad b = \frac{2J}{\sqrt{s'}} \,. \tag{5.8}$$

The parameter $b$ is customarily called the impact parameter.

As extensively discussed in [51, 63], the large-$J$ and fixed-$b$ region is the main source of tension and the search for positive functionals is mainly the search for functionals which are positive on this region. The dispersion relations can have different behaviors in the asymptotic regime (5.8); moreover, some of them assume opposite values in different channels. We summarize the asymptotic behavior of the various equations in table 2. We have

| eq | order | $\rho_J^{(2)}$ | $\rho_J^{(0,+)}$ | $\rho_J^{(0,-)}$ |
|---|---|---|---|---|
| $E_1$ | $O(J^{-6})$ | $0$ | $0$ | $0$ |
| $E_2$ | $O(J^{-6})$ | $0$ | $\frac{H_n^{(0)}(b)}{32}$ | $-\frac{H_n^{(0)}(b)}{32}$ |
| $E_3$ | $O(J^{-6})$ | $\frac{H_n^{(0)}(b)}{64}$ | $\frac{H_n^{(0)}(b)}{64}$ | $\frac{H_n^{(0)}(b)}{64}$ |
| $E_4$ | $O(J^{-6})$ | $-\frac{H_n^{(2)}(b)}{32}$ | $-\frac{H_n^{(2)}(b)}{32}$ | $-\frac{H_n^{(2)}(b)}{32}$ |
| $E_5$ | $O(J^{-6})$ | $0$ | $-\frac{H_n^{(2)}(b)}{32}$ | $\frac{H_n^{(2)}(b)}{32}$ |
| $E_6$ | $O(J^{-4})$ | $-\frac{H_n^{(0)}(b)}{16}$ | $\frac{H_n^{(0)}(b)}{16}$ | $\frac{H_n^{(0)}(b)}{16}$ |
| $E_7$ | $O(J^{-4})$ | $-\frac{H_n^{(2)}(b)}{16}$ | $\frac{H_n^{(2)}(b)}{16}$ | $\frac{H_n^{(2)}(b)}{16}$ |
| $E_8$ | $O(J^{-2})$ | $0$ | $\frac{H_n^{(0)}(b)}{2}$ | $-\frac{H_n^{(0)}(b)}{2}$ |
| $E_9$ | $O(J^{-2})$ | $\frac{H_n^{(0)}(b)}{4}$ | $\frac{H_n^{(0)}(b)}{4}$ | $\frac{H_n^{(0)}(b)}{4}$ |

**Table 2**: Behavior of the equations $E_i$ in the limit $J \to \infty$, $m \to \infty$ with fixed impact parameter $b$, in the three channels: charged, neutral spin-even and neutral spin-odd.

defined

$$H_n^{(k)}(b) \equiv \frac{{}_1F_2\left(\frac{k}{2} + \frac{n}{2} + \frac{1}{2}; \frac{d}{2} - 1, \frac{k}{2} + \frac{n}{2} + \frac{3}{2}; -\frac{b^2}{4}\right)}{k + n + 1} \tag{5.9}$$

and each entry in the last three columns should be multiplied by a the ratio $b/J$ to the appropriate order (second column).[6]

Inspecting table 2 we can see how to combine equations to construct positive functionals: equations such as $E_2$ cannot be considered alone, since there would be no functional that is simultaneously positive in the $\rho^{(0,+)}$ and $\rho^{(0,-)}$ channel. However, it is possible that some combination of $E_3$ and $E_2$ will admit a positive functional. Indeed, as we show later, we can produce positive functionals by considering the 2SDRs alone and thereby obtaining valid bounds on the coefficients entering these relations.

Since two out of three 1SDRs have the asymptotic scaling $O(J^{-6})$, they can be considered together with the 2SDRs. The resulting bounds are expected to be stronger than 2SDRs alone. Equation $E_6$ has a less suppressed behavior: hence, if included, this equation would dominate over the others. However, as for $E_2$, its asymptotics require the same functional to be both positive and negative on the same term. This would force the parameters $a_{6,n}$ of the functional to vanish, rendering the equation useless.

---

[6]A technical note: $E_1$, being a null constraint, has a subleading scaling in this large-$J$ limit compared to $E_2$ and $E_3$, which are also obtained with two subtractions. This seems to be a generic feature of smeared dispersion relations that we have observed in scalars as well as in photon scattering: null constraints are subleading in the regime (5.8). The same happens with fewer subtractions: equations $E_4$ and $E_5$, could be treated as null constraints since they do not contain any new couplings in the low energy part, and we see that they are subleading at large $J$ relative to $E_6$. Similarly, equation $E_7$ can be used as a null constraint; it is suppressed relative to $E_8$ and $E_9$, which contain $g_{0,0}$.

In this case, the $t$-channel dominance assumption introduced in previous sections saves the day: by setting to zero the spectral density $\rho_2$ we do not have to demand positivity of the functional on these terms. Hence, under this assumption, one can also include equation $E_6$.

Finally, 0SDRs $E_{7,8,9}$ have an even less suppressed behavior, but in this case one of the dominant equations does not have this opposite-sign behavior and can safely be used.

In appendix A we discuss a few important technical issues related to the numerical implementation.

In the rest of the section we present the results of our investigations. We addressed the following problems:

1. How do stronger assumptions about Regge behavior, *i.e.* the inclusion of dispersion relations with fewer subtractions, affect the bounds?

   We first answer this question in the most general setup, assuming the convergence of smeared 2SDRs as well as 1SDRs as discussed in [61]. We find that bounds do get stronger but they are not strong enough to restore the positivity bounds that one gets in the forward limit. These bounds are shown in figure 7.

   Next, we assume $t$-channel dominance, which allows us to access the gauge coupling $e^2$ and use an additional 1SDR. We show that, in the limit of zero gauge coupling, certain positivity bounds are restored even in the presence of gravity. The gauge coupling instead introduces another source of negativity. This is shown in figure 9.

2. Are there bounds which involve both the gauge coupling $e^2$ and the gravitational coupling $G$?

   Using the $t$-channel dominance assumption, we produced a positive analytic functional which gives the following inequality

   $$G \leq 10.6618\, e^2 + 0.0367\, g_{0,1}. \tag{5.10}$$

3. Can one impose the Adler's zero condition for Goldstone bosons?

   The non-derivative coupling $g_{0,0}$ can be accessed using 0SDRs. In the presence of forward limit 0SDRs, $g_{0,0}$ must be strictly positive. This directly implies that Goldstone bosons cannot allow for (unsmeared) 0SDRs.

   If we assume the validity of smeared 0SDRs, and in particular $E_9$, then we find that both of the following cannot be true, as they would lead to an inconsistency:

   i) the theory is not coupled to gravity, *i.e.* $G = 0$;

   ii) the scalars are derivatively coupled, *i.e.* $g_{0,0} = 0$, which would be the case for Goldstone bosons.

## 5.3 Adding subtractions

A striking result of [63] was that the existence of the graviton pole relaxes the positivity condition found for certain coefficients in the real scalar $2 \to 2$ amplitude. In section 4 we showed that the coefficient $g_{0,2}$ must be strictly positive when massless exchanges are absent. In presence of massless exchanges this bound is weakened due to the appearance of $1/t$ poles: the graviton pole appears in the 2SDR (5.3), while the photon pole enters the 1SDR (5.4).

A natural question is: can we reinstate the positivity of $g_{0,2}$ by adding constraints which derive from lower subtracted dispersion relation?

In order to address this question we produced bounds on the ratio $g_{0,2}/(8\pi G)$ as a function of the other couplings appearing in 2SDRs: $g_{1,0}$ and $g_{1,1}$. We start by considering equations $E_1, E_2, E_3$, which are the ones derived from 2SDRs. Let us write them in vector notation:

$$
\vec{w}_{\text{high}} = \begin{pmatrix} E_1 \\ E_2 \\ E_3 \end{pmatrix} = \sum_{J \text{ even}} \int_{M^2}^{\infty} ds' \rho_J^{(2)}(s') \, \vec{W}_J^2(s', p) \\
+ \sum_{J \text{ even}} \int_{M^2}^{\infty} ds' \rho_J^{(0,+)}(s') \, \vec{W}_J^+(s', p) + \sum_{J \text{ odd}} \int_{M^2}^{\infty} ds' \rho_J^{(0,-)}(s') \vec{W}_J^-(s', p)
\tag{5.11}
$$

where we have explicitly separated the two neutral channels (spin-even and spin-odd) corresponding to the two positive spectral densities $\rho_J^{(0,\pm)}$ and the charged channel corresponding to $\rho_J^{(2)}$. The corresponding low energy coefficients have the form

$$
\vec{w}_{\text{low}} = 8\pi G \begin{pmatrix} 0 \\ 0 \\ 1/p^2 \end{pmatrix} + g_{1,0} \begin{pmatrix} 0 \\ -1 \\ 0 \end{pmatrix} + g_{1,1} \begin{pmatrix} 0 \\ -p^2 \\ -p^2 \end{pmatrix} + g_{0,2} \begin{pmatrix} 0 \\ 0 \\ 1 \end{pmatrix}
$$
$$
= 8\pi G \vec{w}_G + g_{1,0} \vec{w}_{1,0} + g_{1,1} \vec{w}_{1,1} + g_{0,2} \vec{w}_{0,2} \,.
\tag{5.12}
$$

We restrict to particular smeared functionals (5.6) that annihilate one of the vectors $\vec{w}_{1,0}$ or $\vec{w}_{1,1}$ and obtain bounds on the remaining couplings. The blue region in figure 7 shows the allowed region. As expected the coupling $g_{0,2}$ is allowed to become negative in presence of gravity. We also notice that the two wedges of the allowed region can be approximated by the simple expression

$$
g_{0,2} \geq \frac{1}{2} g_{1,0} - a(8\pi G) \quad \text{and} \quad g_{0,2} \geq -\frac{1}{2} g_{1,0} - b(8\pi G)
\tag{5.13}
$$
$$
\text{with } a \simeq 37.5, \quad b \simeq 24.5 \,.
$$

Hence, when gravity decouples we nicely recover the forward limit bounds.

In order to assess the impact of less subtracted dispersion relations we supplement the 2SDRs with additional equations. Equations $E_4$ and $E_5$ can be added without trouble as they do not introduce any dominant new contribution in the asymptotic regime (5.8), see table 2. Hence we can just enlarge the system (5.11) by adding two more rows. Notice

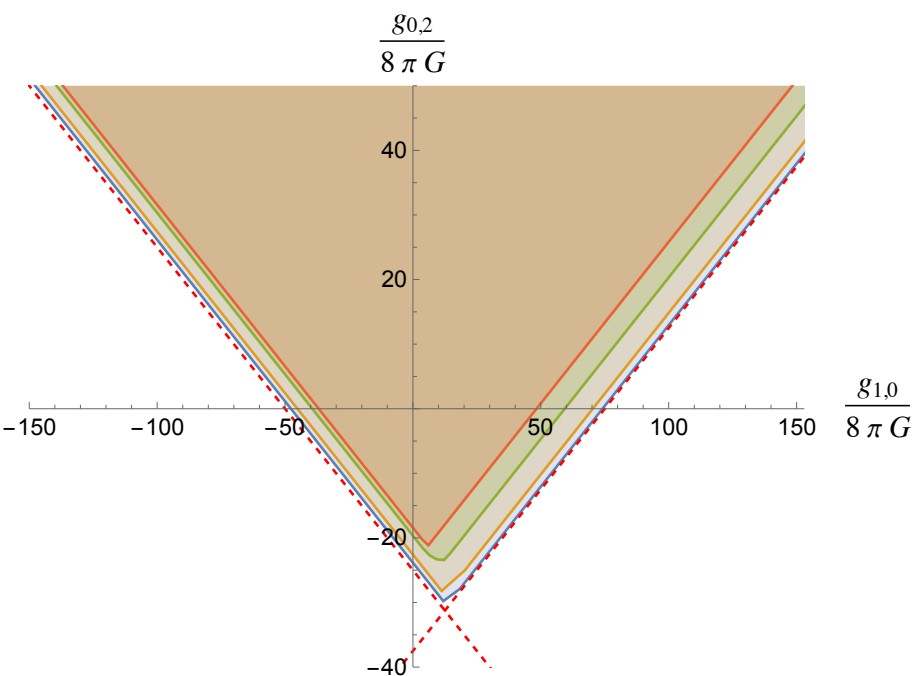

**Figure 7**: Allowed values for the coefficients $g_{1,0}$, $g_{0,2}$ normalized to the Newton constant $8\pi G$ in $d = 5$. From weaker to stronger bounds: only 2SDRs (blue); 2SDRs and 1SDRs with $O(J^{-6})$ asymptotics (orange); 2SDRs with $t$-channel dominance assumption (green); 2SDRs and 1SDRs with $O(J^{-6})$ asymptotics and $t$-channel dominance assumption (red). The slopes of the two wedges are $\pm 1/2$, in agreement with the bounds on the ratio $g_{1,0}/g_{0,2}$ in the forward limit obtained for $d = 4$ in figure 2, which also holds in $d = 5$.

that the low energy part $\vec{w}_{\text{low}}$ will not contain additional vectors as the couplings $g_{0,1}$ and $e^2$ do not appear. We can effectively consider these two new equations as new null constraints. The impact on the bounds of these new equations is also shown in figure 7 and it corresponds to the orange line. Although the bound does become stronger, this is not yet sufficient to restore positivity. A similar analysis is presented for the pair $g_{0,2}$, $g_{1,1}$ in figure 8.

For later convenience, we also inspect the impact of the $t$-channel dominance assumption in this setup. The green and red lines in figure 7 correspond to the bounds obtained using respectively $E_i, i = 1, 2, 3$ only or $E_i, i = 1, \ldots, 5$ supplemented by the assumption $\rho_J^{(0,+)} \equiv 0$. Even in this case $g_{0,2}$ is allowed to be negative.

## 5.4 Recovering positivity from 1SDRs

In order to access the last equation $E_6$, which is the only one that contains the gauge coupling $e^2$, we need to face a new issue. If we consider the asymptotic regime (5.8) as given in table 2, we notice that equation $E_6$ has the leading behavior but unfortunately it is not sign definite. Hence, any functional that is required to be non-negative on this equation, will identically vanish, thus preventing us from getting any bound on the gauge

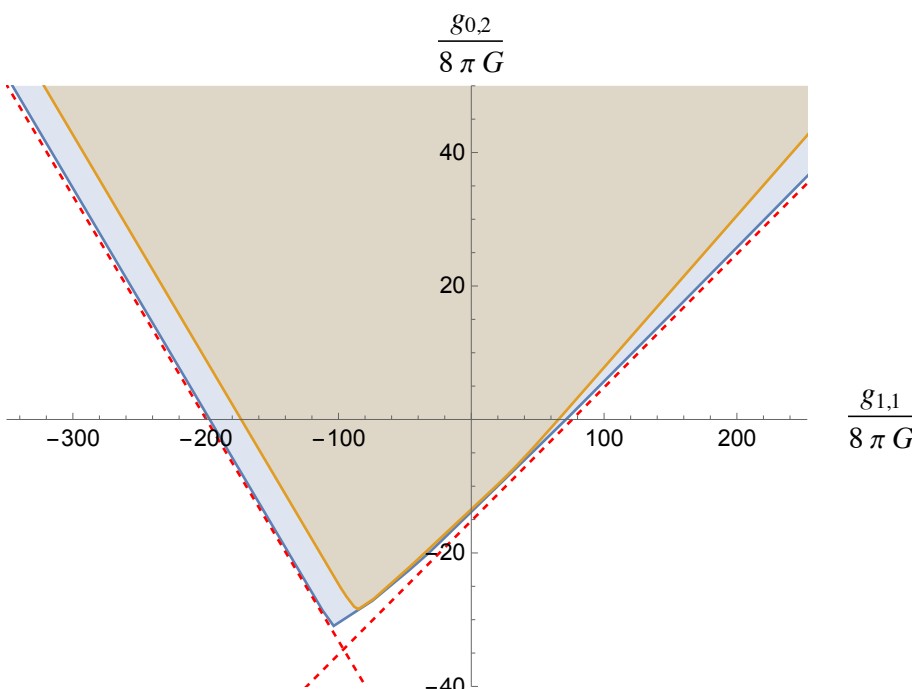

**Figure 8**: Allowed values for the coefficients $g_{1,1}$, $g_{0,2}$ normalized to the Newton constant $8\pi G$ in $d = 5$ dimensions using various dispersion relations and assumptions. From weaker to stronger bounds: only 2SDRs (blue); 2SDRs and 1SDRs with $O(j^{-6})$ asymptotics (orange); 2SDRs with $t$-channel dominance assumption (green); 2SDRs and 1SDRs with $O(j^{-6})$ asymptotics and $t$-channel dominance assumption (red). The slopes of the two wedges are $+1/5$ and $-1/3$, in agreement with the bounds on the ratio $g_{1,1}/g_{0,2}$ in the forward limit obtained for $d = 4$ in figure 2, which also holds in $d = 5$.

coupling.[7]

In order to overcome this difficulty, we specialize to a particular class of theories in which the contribution to the scattering amplitude of the channel $\rho_J^{(2)}$ is parametrically suppressed and can be neglected. This is precisely what happens in the Lovelace-Shapiro amplitude [70, 71] and in pion scattering in the large-$N$ limit [55]. We called this assumption $t$-*channel dominance*, and we have discussed its significance and possible validity more thoroughly in section 4.2.2.

Whenever this is the case, equation $E_6$ becomes sign definite in the asymptotic regime; it still oscillates at large $b$ but we know how to deal with this problem [51, 72].

For the rest of this section, we will work with the $t$-channel dominance assumption and obtain numerical and analytic bounds on the gauge coupling. Let us start with a numerical bound on the same couplings as the previous section. For this analysis we replaced the functional ansatz (5.7) with the following:

---

[7]The same argument applies to $g_{0,1}$.

$$\phi_i(p) = \sum_{n \in S_i} a_{i,n} p^n (1-p)^3 \,. \tag{5.14}$$

This ansatz has a softer behavior near the cut-off and it obeys the assumption of the convergence of smeared 1SDRs in [61]. In this section we only include equations $E_4, E_5, E_6$ for simplicity.

Let us write them in vector notation

$$\vec{v}_{\text{high}} = \begin{pmatrix} E_6 \\ E_5 \\ E_4 \end{pmatrix}$$

$$= \sum_{J \text{ even}} \int_{M^2}^{\infty} ds' \rho_J^{(0,+)}(s') \, \vec{V}_J^+(s',p) + \sum_{J \text{ odd}} \int_{M^2}^{\infty} ds' \rho_J^{(0,-)}(s') \vec{V}_J^-(s',p) \tag{5.15}$$

where we have explicitly separated the two neutral channels (spin-even and spin-odd) corresponding to the two positive spectral densities $\rho_J^{(0,\pm)}$. In $d = 5$ we explicitly we have

$$\vec{V}_J^+ = \begin{pmatrix} \frac{(2J(J+2)-9)p^6 - 9s'p^4}{3s'^4(s'+p^2)} + \frac{{}_2F_1\left(-J,J+2;\frac{3}{2};\frac{p^2}{s'}\right)}{s'^2} \\ \frac{(2J(J+2)-9)p^8 - 2(J(J+2)-6)s'p^6}{3s'^4(s'-p^2)^2} + \frac{p^2(p^2-2s')\,{}_2F_1\left(-J,J+2;\frac{3}{2};\frac{p^2}{s'}\right)}{s'^2(s'-p^2)^2} \\ \frac{(2J(J+2)+21)s'^2p^6 - 4(2J(J+2)-3)s'p^8 + 6(J-1)(J+3)p^{10} - 9s'^3p^4}{3s'^2(s'-p^2)^2(s'+p^2)} + \frac{p^2(p^2-2s')\,{}_2F_1\left(-J,J+2;\frac{3}{2};\frac{p^2}{s'}\right)}{s'^2(s'-p^2)^2} \end{pmatrix} \tag{5.16}$$

$$\vec{V}_J^- = \begin{pmatrix} \frac{(2J(J+2)-9)p^6 - 9s'p^4}{3s'^4(s'+p^2)} + \frac{{}_2F_1\left(-J,J+2;\frac{3}{2};\frac{p^2}{s'}\right)}{s'^2} \\ \frac{(2J(J+2)-9)p^8 - 2(J(J+2)-6)s'p^6}{3s'^4(s'-p^2)^2} + \frac{p^2(2s'-p^2)\,{}_2F_1\left(-J,J+2;\frac{3}{2};\frac{p^2}{s'}\right)}{s'^2(s'-p^2)^2} \\ \frac{2(J(J+2)+3)s'p^6 - 2J(J+2)p^8 - 9s'^2p^4}{3s'^4(s'-p^2)^2} + \frac{p^2(p^2-2s')\,{}_2F_1\left(-J,J+2;\frac{3}{2};\frac{p^2}{s'}\right)}{s'^2(s'-p^2)^2} \end{pmatrix} \tag{5.17}$$

As anticipated we are neglecting the charged channel corresponding to $\rho_J^{(2)}$.

The corresponding low energy coefficients have the form

$$\vec{v}_{\text{low}} = e^2 \begin{pmatrix} \frac{2}{p^2} \\ 0 \\ 0 \end{pmatrix} + G \begin{pmatrix} -5\pi \\ 0 \\ -16\pi \end{pmatrix} + g_{0,1} \begin{pmatrix} 1 \\ 0 \\ 0 \end{pmatrix} + g_{1,0} \begin{pmatrix} -p^2 \\ p^2 \\ 0 \end{pmatrix} + g_{1,1} \begin{pmatrix} p^4 \\ p^4 \\ 2p^4 \end{pmatrix} + g_{0,2} \begin{pmatrix} -2p^2 \\ 0 \\ -2p^2 \end{pmatrix}$$

$$= e^2 \vec{v}_e + G \vec{v}_G + g_{0,1} \vec{v}_{0,1} + g_{1,0} \vec{v}_{1,0} + g_{1,1} \vec{v}_{1,1} + g_{0,2} \vec{v}_{0,2} \,. \tag{5.18}$$

In order to run our numerics it was crucial to restrict to functionals such that $\Lambda[\vec{v}_e] = 1$. This can be done even if the gauge coupling is switched off.[8] Moreover, to obtain bounds

---

[8]Choosing to normalize $\Lambda[\vec{v}_i] = 1$ for any other $\vec{v}_i \neq \vec{v}_e$ does not lead to any bound: it would be nice to understand the technical reason for this.

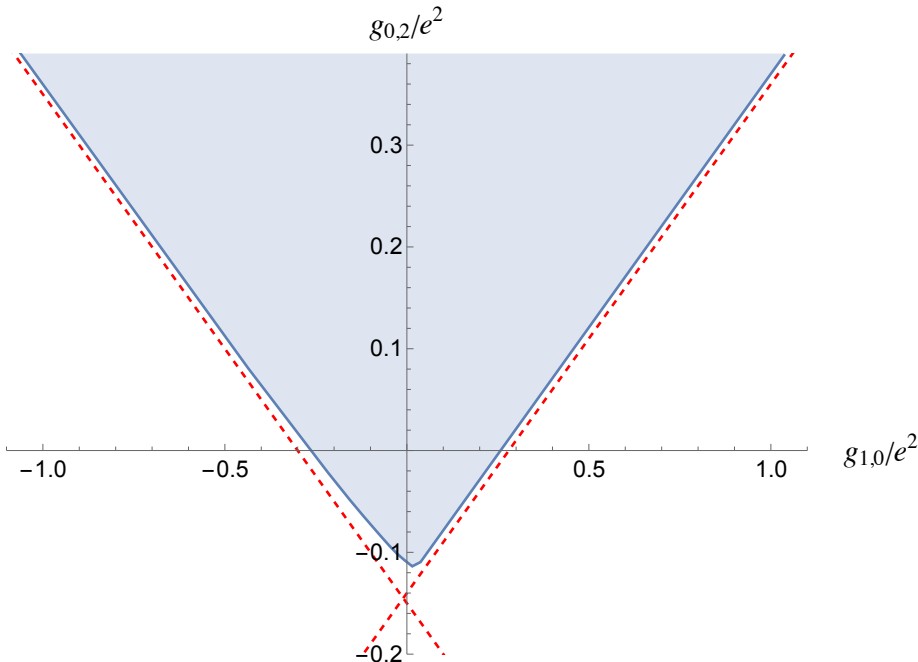

**Figure 9**: Allowed values for the coefficients $g_{1,0}$, $g_{0,2}$ normalized to the gauge constant $e^2$ in $d = 5$ dimensions. The shaded region is allowed, assuming 1SDR and $t$-channel dominance.

independent on the other couplings we can also demand

$$\Lambda[\vec{v}_{0,1}] = \Lambda[\vec{v}_{1,1}] = \Lambda[\vec{v}_G] = 0 \tag{5.19}$$

$$\text{or} \quad \Lambda[\vec{v}_{1,0}] = \Lambda[\vec{v}_{1,1}] = \Lambda[\vec{v}_G] = 0 \tag{5.20}$$

$$\text{or} \quad \Lambda[\vec{v}_{1,0}] = \Lambda[\vec{v}_{0,1}] = \Lambda[\vec{v}_G] = 0 \,. \tag{5.21}$$

In all cases above we will obtain bounds independent of $G$, which therefore hold both with and without gravity.[9]

Let us take the conditions (5.19) as an example. A positive functional on the RHS of (5.18) will lead to a condition

$$e^2 + g_{0,2}\Lambda[\vec{v}_{0,2}] + g_{1,0}\Lambda[\vec{v}_{1,0}] \geq 0 \tag{5.22}$$

which translates to an excluded region in the plane $(g_{1,0}/e^2, g_{0,2}/e^2)$. We have numerically computed the optimal such bound in figure 9. Hence we conclude that as long as the gauge coupling is non-vanishing, a little negativity for the coupling $g_{0,2}$ is still consistent with the bootstrap constraint.

The allowed region can be approximated by the following conditions

$$g_{0,2} \geq \frac{1}{2}g_{1,0} - \tilde{a}e^2 \quad \text{and} \quad g_{0,2} \geq -\frac{1}{2}g_{1,0} - \tilde{b}e^2 \tag{5.23}$$

with $\tilde{a} \simeq 0.13, \quad \tilde{b} \simeq 0.15 \,.$

---

[9]In the latter case the condition $\Lambda[\vec{v}_G] = 0$ could be dropped and we would have more general functionals available.

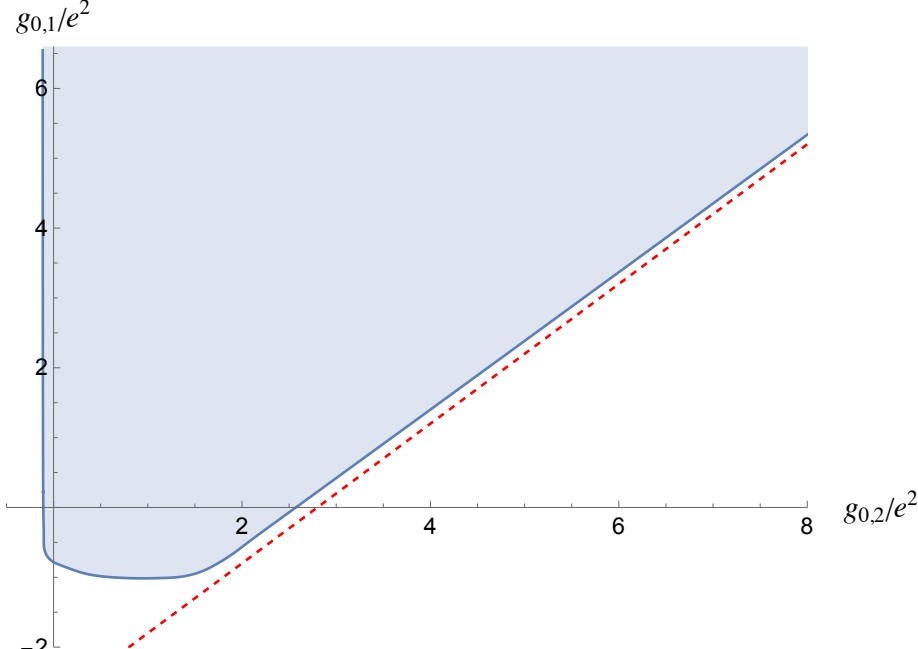

**Figure 10**: Allowed values for the coefficients $g_{0,2}$, $g_{0,1}$ normalized to the gauge constant $e^2$ in $d = 5$ dimensions. The shaded region is allowed, assuming 1SDR and $t$-channel dominance.

However, since we chose functionals which are applicable for any value of the coefficients, these bounds lead to an interesting conclusion: in the limit $e^2 \to 0$, the positivity bound $g_{0,2} \pm \frac{1}{2}g_{1,0} \geq 0$ found in the forward limit is restored, even in presence of gravity! In section 5.6 we give a weaker (but analytic) analog of these bounds.

In figure 10 we show the allowed region for $(g_{0,1}/e^2, g_{0,2}/e^2)$. In this case one finds that $g_{0,1} \geq 0$ in the limit of vanishing gauge coupling. A similar plot can also be obtained for $g_{1,1}$ and $g_{0,2}$, although we do not show it here.

## 5.5 Analytic bounds for $e^2$ and $G$

In the previous section we obtained numerical bounds independent of the gravity coupling $G$. Here instead we want to obtain bounds involving both $e^2$ and $G$. We will not try to get the best bound, but rather we will provide a functional, which can be shown to be positive by inspection and use it to show that whenever the $t$-channel dominance assumption holds, in presence of gauge coupling and gravity, one has the inequality

$$G \leq 10.6618\, e^2 + 0.0367\, g_{0,1} \,. \tag{5.24}$$

The above constraint has deep and intriguing consequences. When phrased as a bound on $G$, it is a form of the weak gravity conjecture: the strength of gravity is bounded by the gauge force plus the self-couplings.[10] It is also very interesting that for $g_{0,1} < 0$, we cannot

---

[10]We have tried to find a bound on $G$ and $e^2$ which does not involve any other couplings but we have not succeeded. However it is interesting that we find such a weak dependence on $g_{0,1}$. Perhaps it would be

turn off the gauge coupling and maintain a positive Newton's constant. This is reminiscent of the conjecture that there are no global symmetries in the presence of gravity: gauged $U(1)$ becomes global in the $e^2 \to 0$ limit. For $g_{0,1} < 0$ and $e^2 = 0$, we see that our bound also requires that $G \to 0$.

Given the importance of this statement we formulate it as a theorem:

## Theorem

Consider the scattering amplitude of a complex scalar in $d = 5$, coupled to gravity and a gauge field associated to the $U(1)$ symmetry that rotates the complex field.[11] Assume that the theory is weakly coupled below a cutoff $M$, (which we set to 1 for compactness) and the low energy amplitudes are given by (3.3). Above the cutoff the amplitudes are generic and admit a partial wave decomposition as in (3.8). Given an amplitude $\mathcal{A}$, Consider the smeared amplitudes

$$\widetilde{\mathcal{A}}_\phi(s) \equiv \int_0^1 \phi(p)\mathcal{A}(s, -p^2)dp\,. \tag{5.25}$$

Let us assume that

$$\lim_{s \to \infty} \frac{\widetilde{\mathcal{A}}_\phi(s)}{s} = 0 \qquad \text{for any } \phi(p) \text{ such that } \begin{cases} \lim_{p \to 0} p^{-2}\phi(p) < \infty \\ \lim_{p \to 1}(1-p)^{-3}\phi(p) < \infty \end{cases} \tag{5.26}$$

for $\mathcal{A} = \mathcal{A}^{++++}, \mathcal{A}^{+--+}, \mathcal{A}^{+-+-}$. These assumptions have been proven to follow from basic assumptions on scattering amplitudes in the presence of gravity [61].

In addition we assume $t$-channel dominance: $\rho_J^{(2)}(s) = 0$ for $s \geq M^2$.

Then, there exists a bound of the form

$$G \leq Ae^2 + Bg_{0,1}, \qquad \text{with } A, B > 0 \tag{5.27}$$

and numerically given in $d = 5$ by (5.24).

## Proof

Let us now consider the following functionals acting on a three dimensional vector $\vec{v}$ of functions of $p$:

$$\Lambda_1[\vec{v}] = \int_0^1 \frac{p^2(1-p)^3}{10}\left(10v_1(p) + 18pv_2(p) - (5+9p)v_3(p)\right)dp, \tag{5.28}$$

$$\Lambda_2[\vec{v}] = \int_0^1 p^3(1-p)^3(v_1(p) + v_2(p) - v_3(p))dp, \tag{5.29}$$

$$\Lambda_3[\vec{v}] = \int_0^1 \frac{p^2(1-p)^3}{10}(5-9p)(2v_2(p) - v_3(p))dp. \tag{5.30}$$

---

possible to study the behavior as the number of constraints increases to argue that asymptotically there is a bound not involving $g_{0,1}$.

[11]We believe that a similar theorem should hold in $d > 5$, with minor modifications on the assumptions.

It is straightforward to show that these functionals annihilate all low energy vectors in (5.18) except $\vec{v}_e$, $\vec{v}_G$ and $\vec{v}_{0,1}$. Then we will be interested in larger functionals taking the form

$$\Lambda[\vec{v}] = a_1 \Lambda_1[\vec{v}] + a_2 \Lambda_2[\vec{v}] + a_3 \Lambda_3[\vec{v}] \,. \tag{5.31}$$

The positivity properties can be visualized by plotting the functionals: for each term appearing in the sum (5.15), one can compute the result of acting with $\Lambda_1$, $\Lambda_2$ and $\Lambda_3$ and plot them as entries of a three dimensional vector (with suitable normalization). For each value of $J$ one obtains a curve parametrized by $s'$. The existence of a positive functional is equivalent to the statement that all the curves lie on the same side of a half-space. A careful inspection reveals that such a plane exists and it is fixed by the following two conditions:

$$\Lambda[\vec{V}_1^-(s'=1,p)] = 0, \tag{5.32}$$

$$\Lambda[\vec{V}_{J\to\infty}^+(s'=1,p)] = 0. \tag{5.33}$$

In our language, a positive functional $\Lambda \sim \{a_1, a_2, a_3\}$ corresponds indeed to (the vector normal to) the plane that leaves all the curves on the same half space, and a plane is uniquely identified by two vectors.

The above conditions uniquely fix $\Lambda$ up to a normalization. We have numerically verified that this choice of $a_1, a_2, a_3$ makes the action of $\Lambda$ on $\vec{v}_{\text{high}}$ non-negative and it approximately corresponds to

$$(a_1, a_2, a_3) = (1., -2.19807, 0.376782) \,, \tag{5.34}$$

though to maintain positivity of the functional, it is important to keep hundreds of digits of precision.

For a better visualization we show, in figure 11, a two dimensional projection of the three dimensional space. More precisely, given the unit-normalized vectors $\hat{n}_i$

$$\hat{n}_1 = \mathcal{N}_1 \begin{pmatrix} \Lambda_1[\vec{V}_1^-(1,p)] \\ \Lambda_2[\vec{V}_1^-(1,p)] \\ \Lambda_3[\vec{V}_1^-(1,p)] \end{pmatrix} \,, \qquad \hat{n}_2 = \mathcal{N}_2 \begin{pmatrix} \Lambda_1[\vec{V}_{J\to\infty}^+(1,p)] \\ \Lambda_2[\vec{V}_{J\to\infty}^+(1,p)] \\ \Lambda_3[\vec{V}_{J\to\infty}^+(1,p)] \end{pmatrix} \tag{5.35}$$

and $\hat{n}_3$ such that: $\hat{n}_3 \cdot \hat{n}_1 = \hat{n}_3 \cdot \hat{n}_2 = 0$

we plot the combinations

$$\hat{x} = \hat{n}_3 + \frac{\hat{n}_1 - \hat{n}_1}{\sqrt{2}} \,, \qquad \hat{y} = \hat{n}_3 - \frac{\hat{n}_1 - \hat{n}_1}{\sqrt{2}} \,. \tag{5.36}$$

Let us identify the relevant features of this plot. The green line represents the asymptotic regime (5.8) and it is parametrized by the impact parameter $b \geq 0$. The point $b = 0$ identifies a direction to which all other curves must asymptote for fixed $J$ and large $m$. At large $b$ we see oscillations. The other colored lines correspond to the action of the functionals on $V_J^\pm$ for even/odd $J$. We only show complete curves for a few values of $J$ and the starting point of more curves for larger values of $J$, shown as arrows. Finally, the dashed red line divides the plane in two half-spaces, leaving all curves on the same side: it represents the positive functional. As expected by construction, this line is saturated by the direction corresponding to $J = 1, s' = 1$ and the direction $J \to \infty, s' = 1$. The action of this functional on (5.18) produces the bound (5.24).

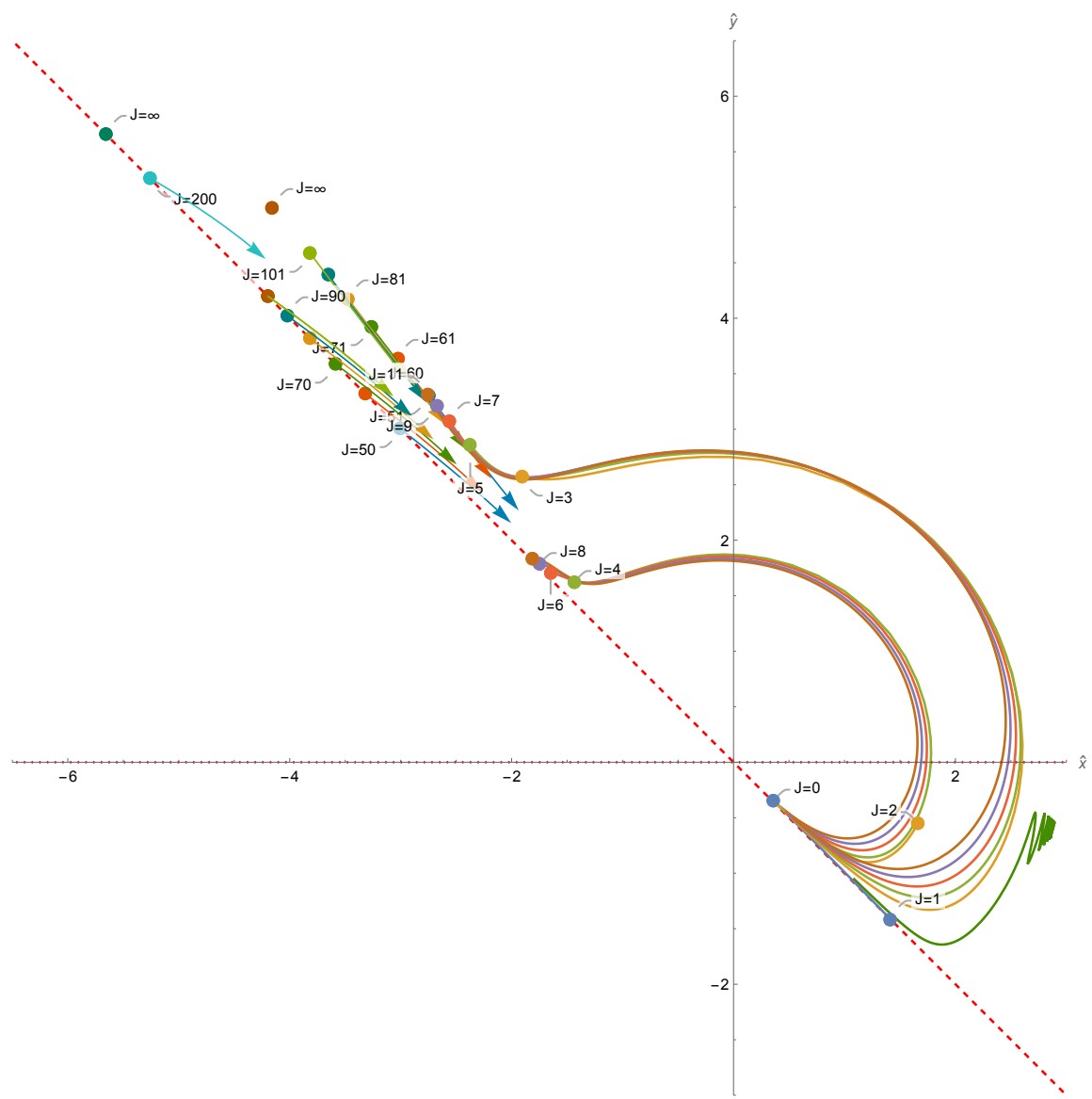

**Figure 11**: Curves in the plane $(\hat{x}, \hat{y})$ defined in (5.36). This is essentially a two-dimensional projection of the functionals defined in (5.31). The fact that all points lie in the same half of the plane implies that their sum is positive (when dotted with an appropriate vector), which in turn implies that a corresponding low-energy combination is positive via the sum-rules. This leads to the bound (5.24). See the text for a full description.

## 5.6  More analytic bounds from 1SDRs

Following similar steps as in the previous section, one can prove more inequalities among EFT coefficients. Consider for instance the functional

$$\Lambda[\vec{v}] = b_1 \Lambda_1[\vec{v}] + b_2 \Lambda_2[\vec{v}] + b_3 \Lambda_3[\vec{v}] \tag{5.37}$$

where this time the $\Lambda_i$ are given by

$$\Lambda_1[\vec{v}] = \int_0^1 \frac{p^2(1-p)^3}{1280} \left(1280v_1(p) + 2304v_2(p) + (1607 - 4683p)v_3(p)\right) dp, \tag{5.38}$$

$$\Lambda_2[\vec{v}] = \int_0^1 \frac{p^2(1-p)^3}{256} (p(256v_1(p) + 256v_2(p) - 619v_3(p)) + 231v_3(p)) dp, \tag{5.39}$$

$$\Lambda_3[\vec{v}] = \int_0^1 \frac{p^2(1-p)^3}{20} (4(9p - 5)v_2(p)(7p - 3)v_3(p)) dp. \tag{5.40}$$

One can obtain positive functionals by requiring

$$\Lambda[\vec{V}_1^-(s' = 1, p)] = 0, \tag{5.41}$$

$$\Lambda[\vec{V}_{J\to\infty}^+(s' = 1, p)] = 0. \tag{5.42}$$

The resulting functional, approximately given by

$$(b_1, b_2, b_3) = (1., -2.20608, 0.418084), \tag{5.43}$$

produces the bound

$$g_{0,2} \geq -3467.19e^2 - 11.28g_{0,1}. \tag{5.44}$$

Alternatively one can consider the coefficients identified by the conditions

$$\Lambda[\vec{V}_1^-(s' = 1, p)] = 0, \tag{5.45}$$

$$\Lambda[\vec{V}_2^+(s' = 1, p)] = 0. \tag{5.46}$$

This time the resulting functional, approximately given by

$$(b_1, b_2, b_3) = (1., -2.06745, 4.45382), \tag{5.47}$$

produces the bound

$$g_{0,2} \leq 223.166e^2 - 1.445g_{0,1}. \tag{5.48}$$

In a theory with $e^2 = 0$ these equations imply $g_{0,1} \geq 0$, regardless of the presence of gravity (the functional annihilates G). If we now go back to (5.24) we see that without gauge interactions $G$ is bounded by $g_{0,1}$. When the theory is gauged, $g_{0,1}$ is allowed to be negative, but still bounded from below by $e^2$.

## 5.7 Analytic bounds from 0SDRs

We could apply a similar reasoning to 0SDRs. Here for simplicity we can consider only equations $E_7, E_8, E_9$:

$$\vec{z}_{\text{high}} = \begin{pmatrix} E_9 \\ E_8 \\ E_7 \end{pmatrix} = \sum_{J \text{ even}} \int_{M^2}^\infty ds' \rho_J^{(2)}(s') \, \vec{Z}_J^+(s', p)$$

$$+ \sum_{J \text{ even}} \int_{M^2}^\infty ds' \rho_J^{(0,+)}(s') \, \vec{Z}_J^+(s', p) + \sum_{J \text{ odd}} \int_{M^2}^\infty ds' \rho_J^{(0,-)}(s') \vec{Z}_J^-(s', p). \tag{5.49}$$

The corresponding low energy coefficients have the form

$$\vec{z}_{\text{low}} = g_{0,0}\begin{pmatrix} 1 \\ 1 \\ 0 \end{pmatrix} + e^2\begin{pmatrix} -2 \\ 2 \\ 2 \end{pmatrix} + G\begin{pmatrix} 5\pi p^2 \\ 19\pi p^2 \\ 3\pi p^2 \end{pmatrix} + g_{0,1}\begin{pmatrix} -p^2 \\ p^2 \\ p^2 \end{pmatrix} + g_{1,0}\begin{pmatrix} 0 \\ 0 \\ p^4 \end{pmatrix} + g_{0,2}\begin{pmatrix} p^4 \\ p^4 \\ -p^4 \end{pmatrix}$$
$$= g_{0,0}\vec{z}_{0,0} + e^2\vec{z}_e + G\vec{z}_G + g_{0,1}\vec{z}_{0,1} + g_{1,0}\vec{z}_{1,0} + g_{0,2}\vec{z}_{0,2}\,. \tag{5.50}$$

Consider now the functional

$$\Lambda[\vec{z}] = \sum_{i=7}^{8}\int_0^1 (f_{i,2}p^2 + f_{i,3}p^3)(1-p)^3 z_i(p)dp \tag{5.51}$$

with the choice:

$$f_{9,2} = 1, \qquad\qquad f_{8,2} = \frac{1}{2}, \qquad\qquad f_{7,2} = \frac{1}{2},$$
$$f_{9,3} = -\frac{11}{7}, \qquad\qquad f_{8,3} = -\frac{11}{14}, \qquad\qquad f_{7,3} = -\frac{11}{14}\,. \tag{5.52}$$

This choice implies

$$\Lambda[\vec{z}_{0,1}] = \Lambda[\vec{z}_{1,0}] = \Lambda[\vec{z}_{0,2}] = \Lambda[\vec{z}_{e^2}] = 0 \tag{5.53}$$

so that the bound produced by $\Lambda$ does not depend on the value of those couplings. One can show that with the above choice one has $\Lambda[\vec{Z}_J^2(s',p)] = 0$ for all $J$ odd, while the action on any other vector appearing in the high energy partial wave expansion is positive. In the end, acting on the low energy (5.50) we get

$$g_{0,0} \geq -\frac{1}{9}G\,. \tag{5.54}$$

Notice that the bound is independent of the gauge coupling $e^2$. The above relation implies that even in the case of smeared functionals, in absence of gravity $g_{0,0}$ must be non-negative. As mentioned, a careful inspection of the functional reveals that it annihilates exactly all odd spins, while it is strictly positive on the rest. Hence a putative amplitude with vanishing $g_{0,0}$, if the 0SDRs has to hold, could only contain odd spins. This is not the case for most known amplitudes of Goldstone bosons, which therefore must not admit the 0SDRs we used. It is still plausible that 0SDRs exist, but only when smeared with kernels of the form $p^n(1-p)^k$ with $n, k$ sufficiently high.[12]

## 6  Discussion

In this paper, we have examined the positivity bounds on an EFT of charged scalars. The goal was to determine how varying the number of subtractions affected the bounds. The general intuition, that having fewer subtractions would lead to stronger bounds, was true

---

[12]On the other hand, restricting to $n, k$ too large makes it impossible to find positive functionals in the asymptotic regime (5.8).

when going from 2 to 0 subtractions, but we found that going from 2 to 1 subtraction did not lead to significantly stronger bounds. However, we considered another assumption: that $\rho_J^{(2)}(s)$ is zero for $s$ positive and real. This assumption, equivalent to the idea that no $s$-channel diagrams contribute to the $\rho_J^{(2)}(s)$, was termed $t$-channel dominance.

We studied the complex scalar system with a number of different low-energy assumptions (namely, the presence or absence of massless states) and high-energy assumptions, (the Regge behavior and $t$-channel dominance). Our findings include:

- 1SDR is not significantly stronger than 2SDR, but 1SDR + $t$-channel dominance is.

- In the absence of massless exchanges, our bounds for 1SDR + $t$-channel dominance are identical to those explored for pion scattering in [53, 55].

- Assuming 1SDR + $t$-channel dominance, plus $e^2 \to 0$, restores positivity for $g_{0,2}$, which is positive in the forward limit but allowed to be negative in the presence of gravity. We conclude that if $g_{0,2}$ is negative, then the $U(1)$ symmetry of the complex scalar must be gauged.

- Assuming 1SDR and $t$-channel dominance, we find a bound on the strength of the gravitational coupling:

$$G \leq 10.6618e^2 + 0.0367g_{0,1} \tag{6.1}$$

in units of the cutoff scale $M$.

Several of our results seem morally related to certain ideas on symmetries in quantum gravity. The bound on $G$ is very similar to the weak gravity conjecture, and $g_{0,2} < 0 \implies$ "$U(1)$ must be gauged" is at least reminiscent of the no-global-symmetry conjecture. It would be quite nice to extend this work to the scattering of massive particles. In general, we believe that adding a mass should not change EFT bounds much as long as the mass is small compared to the cutoff scale $M$. However in this particular case, it would be interesting to look at the allowed region when the mass is larger or smaller than the electric charge. Adding a mass will introduce some technical difficulties but it would make for a much cleaner comparison of our bounds with the weak gravity conjecture. In general, it is our opinion that using dispersion relations as a "bottom-up" method to consider the Swampland conjectures is an interesting and important path that we hope to continue considering in the future.

This work presented a puzzle, which applies even for the case of the real scalar: what is the fate of amplitudes which marginally violate the Regge-boundedness assumptions? In particular, for the real scalar, we saw that with a 0-subtractions, the heavy scalar amplitude

$$A_{\text{scalar}} = \frac{1}{m^2 - s} + \frac{1}{m^2 - t} + \frac{1}{m^2 - u} \tag{6.2}$$

goes like $s^0$ at large-$s$ and so technically it does not satisfy the bound $\lim_{s \to \infty} \mathcal{A} < s^0$. However it remains in the allowed region when we assume the validity of 0SDRs. On the other hand, there are amplitudes of the form $A_{stu-\text{pole}} - \gamma A_{\text{scalar}}$, which have all positive

spectral densities (they must because they are in the 2SDR allowed region) and they also have $s^0$ behavior at large-$s$. However they are disallowed by the 0SDR bounds. This seems quite strange to us– we believe that it has to do with whether the amplitudes can be "fixed"– whether there is a small change which will infinitesimally soften the Regge behavior without violating positivity. It would be great to make this notion precise, or more generally to understand the conditions under which marginal amplitudes are allowed.

Another interesting direction would be to investigate what happens with other low energy poles. For example, if we assume that the first massive state is a scalar with mass $M_1$ and then impose a further cutoff at $M_2$. This has been suggested as a potentially promising route to studying large-$N$ pion scattering by including in the low-energy the effects of tree-level rho-meson exchange and higher spin states [73]. We think that, even in the case of a single real scalar, there is a lot to be explored by varying the mass of the first excited states and its coupling.

Finally, we should return to the issue of $t$-channel dominance, which led to virtually all of the interesting results we obtained in this paper. There is more to do in understanding when the condition is valid. In particular, it seems to require that there is no (or only a very weak) coupling to charge-two states, as these would contribute to $\rho^{(2)}$ at tree-level. This condition, as we remarked in section 4, is morally related to the weak gravity and repulsive force conjectures: if the scalar field has mass greater than charge, then it should form a charge-two bound state which can contribute to $\rho^{(2)}$. If the scalar has mass less than charge, satisfying the conjectures, then no such bound state will exist to contribute to $\rho^{(2)}$. More generally the requirement is that the coupling to charge-two states is much weaker than the coupling to neutral states, so that $\rho^{(2)} \ll \rho^{(0,\pm)}$. It would be interesting to see if there is some setting, like the CFT bootstrap, where this assumption holds or is violated. One might also see if the converse holds – could requiring WGC in some form like $G < c_1 e^2$ imply some for of $t$-channel dominance, $\rho < c_2$, for some constants $c_1$ and $c_2$? This question could potentially be addressed numerically – $e^2$ and $G$ could be fixed and then one could try to maximize $\rho_2$ at various values of $s$ or maximize some of its moments. This would be very interesting to try.

The assumption of $t$-channel dominance may also require that the UV completion is weakly coupled; otherwise $\rho^{(2)}$ will get significant contributions from 1-loop diagrams which are unavoidable when coupling to photons or gravitons. The EFT bounds derived in this paper do not in general require a weakly-coupled UV completion. Nonetheless it is interesting to note that currently every known example of a completion saturating the EFT bounds comes from integrating out a particle at tree-level. Thus it seems that the methods used here are ideal for studying weakly-coupled UV completions.

Finally, the fact that $t$-channel dominance seems to be required to effectively use 1SDRs to improve positivity bounds inspires speculation on the connection between the two. It is interesting to note that both seem to be valid in the case of pion scattering at large $N$. It would be interesting to understand if $t$-channel dominance implies a stronger Regge behavior in any setting. It would also be interesting to understand the validity of $t$-channel dominance in the presence of gravity. In fact, it was the validity of (smeared) 1SDRs for gravitational scattering [61] that inspired the present work in the first place.

Perhaps this could be aided using extra assumptions about gravitational scattering, such as the formation of black holes at small impact parameter. Another interesting question is to understand if adding 1SDRs + $t$-channel dominance will restore positivity in other important cases. For example, one could extend the analyses of [51] to higher dimension, using the formalism developed in [57, 65], and hope that $t$-channel dominance + 1SDRs leads to proving the black holes formulation of the weak gravity conjecture [74].

## Acknowledgments

We would like to thank Jan Albert, Leonardo Rastelli, Sasha Zhiboedov, Johan Henriksson and Simon Caron-Huot for interesting conversations and comments on this work. MV would also like to thank SISSA for the availability of the Ulysses cluster, as well as Scuola Normale Superiore for providing the machines that were used in the early part of this work. Some computations were run on the INFN-PISA cluster.

This work has received funding from the European Research Council (ERC) under the European Union's Horizon 2020 research and innovation program (grant agreement no. 758903).

## A    Details on numerics

In this section we provide few additional technical details about the numerical implementation of the algorithm discussed in the main text.

The determination of bounds in the forward limit was carried out following the algorithm developed in [40]. To obtain the desired bounds, we carefully selected pertinent sum rules and complemented them with a series of null constraints derived from $k$-subtracted dispersion relations. Positivity conditions were enforced for a finite set of spins, each satisfying $s' \geq M^2$. In addition, in order to ensure positivity for all spins and masses, we included vectors in the limit (5.8). We consider a generic vector of partial waves and made the replacement $s' \to 4J^2/b^2$ and took the leading power in $J$. The resulting vector, modulo a prefactor, is a polynomial in $b$ and can be fed to SDPB as usual.
In table 3 we collect all the relevant information.

Next let us discuss the bounds on section 5, using smeared functionals. We followed the same procedure described in appendix C of [51] and we report in table 4 our choices of parameters.

The central point of discussion revolves around the choice of normalization for the functional. Regrettably, we have not discovered an efficient method to incorporate the subleading behavior outlined in (5.8) for each equation. Consequently, our approach has been limited to imposing positivity constraints on a finite set of spins, which includes instances of sparse large values, as well as on the leading asymptotic term.

We've noticed that the selection of the normalization vector plays a crucial role when employing $k$-subtracted dispersion relations. It is essential for this vector to correspond to a coupling that can only be probed with $k$ subtractions or fewer. Failing to adhere to this criterion would lead SDPB to opt for a functional that vanishes when applied to the

asymptotic behavior described in (5.8). Such a functional, which annihilates the leading term, would prove excessively stringent, particularly resulting in negative outcomes when tested with spins not included in the numerical analysis. This, in turn, could artificially yield overly strict bounds or even lead to complete exclusion.

| Fig. | $k$SDR | spin range | spins imposed | # null c. |
|------|--------|------------|---------------|-----------|
| 1a, 1b | 0 | $J \geq 0$ | R | 45 |
| 1a, 1b | 2 | $J \geq 0$ | R | 30 |
| 2a, 2b | 2 | $J \geq 0$ | A | 10 |
| 2a, 2b | 1 | $J \geq 0$ | A | 15 |
| 2a, 2b | 0 | $J \geq 0$ | A | 28 |
| 3a, 3b | 2 | $J = 0, 1$ | $J = 0, 1$ | 0 |
| 3a, 3b | 2 | $J \geq 0$ | A | 10 |
| 3a, 3b | 2 | $J \geq 2$ | B | 10 |
| 6a | 1 | $J \geq 0$ | C | 105 |
| 6a | 1 | $J = 0$ | $J = 0$ | 0 |
| 6a | 1 | $J \geq 1$ | D | 105 |
| 6b | 1 | $J \geq 0$ | C | 105 |
| 6b | 1 (large $N$) | $J \geq 0$ | C | 105 |
| 6b | 0 | $J \geq 0$ | C | 231 |
| 6b | 0 (large $N$) | $J \geq 0$ | C | 231 |

where the spin ranges are defined as

| | $J$ even | $J$ odd |
|---|----------|---------|
| R | $\{0, 2, \ldots, 60, 70, \ldots, 150\}$ | $-$ |
| A | $\{0, 2, 4, \ldots, 100\}$ | $\{1, 3, 5, \ldots, 101\}$ |
| B | $\{2, 4, \ldots, 100\}$ | $\{3, 5, \ldots, 101\}$ |
| C | $\{0, 2, 4 \ldots, 40\}$ | $\{1, 3, 5 \ldots, 41\}$ |
| D | $\{2, 4 \ldots, 40\}$ | $\{1, 3, 5 \ldots, 41\}$ |

**Table 3**: Number of null constraints and spin ranges used in the various forward plots. Further notice that in all plots in the table the limit $J \to \infty$, $m$ fixed has been included, while in the last plot of the list we have also included the $J \to \infty$, $b$ fixed limit.

As mentioned, this can be prevented by normalizing the functional in the proper way:

- for 2SDRs we demanded $\Lambda[\vec{w}_G] = 1$, *i.e.* the term multiplying $G$ in (5.12).

- the same condition holds when including 1SDRs that do not alter the leading asymptotics.

- for all 1SDRs we demanded $\Lambda[\vec{v}_e] = 1$, *i.e.* the term multiplying $e^2$ in (5.18).

- for 0SDRs we demanded $\Lambda[\vec{z}_{0,0}] = 1$, *i.e.* the term multiplying $g_{0,0}$ in (5.50).

| Fig. | eqns | functional | # f.n.c. | Spins |
|:---:|:---:|:---:|:---:|:---:|
| 7 (blue/green) | $E_1, E_2, E_3$ | $p^n(1-p),$ $n = 2, 3 \ldots 11$ | 15 | $\mathcal{S}_J$ |
| 7 (red/orange) | $E_1, E_2, E_3,$ $E_4, E_5$ | $p^n(1-p),$ $n = 2, 3 \ldots 11$ | 15 | $\mathcal{S}_J$ |
| 8 (blue) | $E_1, E_2, E_3$ | $p^n(1-p),$ $n = 2, 3 \ldots 11$ | 15 | $\mathcal{S}_J$ |
| 8 (orange) | $E_1, E_2, E_3,$ $E_4, E_5$ | $p^n(1-p),$ $n = 2, 3 \ldots 11$ | 15 | $\mathcal{S}_J$ |
| 9,10 | $E_4, E_5, E_6$ | $p^n(1-p)^3,$ $n = 2, 3 \ldots 7$ | / | $\mathcal{S}_J$ |

**Table 4**: Choices of equations, functionals and parameters used in the figures presented in section 5. The fourth column indicates the number of forward null constraints obtained with more than 2 subtractions. The spin set used is: $\mathcal{S}_J = \{0, 1, \ldots, 40\} \cup \{50, 51, 60, 61, \ldots, 201\}$.

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
