# Peer review of "Adding subtractions: comparing the impact of different Regge behaviors"

_SciPost Physics_

## Round 1 · Author Response

We thank both reviewers for their feedback and suggestions.

---

## Round 1 · List of Changes

Reviewer 1

(1) We did construct all of the amplitudes that appear in the plot, and indeed the amplitude that the reviewer suggests is proportional to the amplitude in 2.11. Possibly this is unclear since we originally called that amplitude A_stu instead of A_stu-pole. We have changed this for clarity.

(2) This is a good question that we wondered about, but we don’t know. We guess that the answer is yes and we made a few other plots internally that we didn’t include in the paper that had e.g. spin 4 and spin 6 extremal. We added a brief speculation about this at the very end of section 2

(3) Single spectral density means that all spectral densities are zero except one. In general, there is only a single amplitude that is 1) single spectral density and 2) (up to addition of contact terms and rescaling by a constant). This probably isn’t particularly deep but we wanted to point it out anyways. We also elaborated that these are extremal in the sense that all of the spectral densities except one have been dialed to their minimum value, which is zero. The value of the remaining spectral density doesn’t matter since these bounds only care about ratios. We have rephrased the paragraph to be more clear.

(4) This was a typo – we meant section 5.2, which is where we elaborate. This paragraph is meant to summarize the issue with the null constraints from 0SDR in the context of pion scattering. The forward limit 0SDRs imply g_00 > 0 but pions are goldstone bosons so they must have g_00 = 0 (they’re derivatively coupled). We’ve fixed the typo and rephrased the paragraph.

(5) We added that these figures were made assuming 1SDR and t-channel dominance.

Reviewer 2:

(1) We see the reviewer’s point here. The fact that g_01 can’t be bounded is discussed below equation 4.5. This is primarily what the line in the intro referred to. But the fact that these null constraints are useless in general is more of an empirical fact – including 1SDR constraints (without t-channel dominance) led to identical plots as when we didn’t include them. We have changed the wording and added a footnote on page 3 about how they might be useful in the non-linear context.

(2) That footnote was misleading – it is meant to say that normalizing any other Lambda[v_i] other than Lambda[v_e] does not lead to a bound, not that the actual value of the normalization matters. We have reworded this to clarify.

(3) No, this doesn’t violate crossing, at least not the crossing symmetry that we’ve assumed for the complex scalar (which is t-u crossing symmetry). To see this, consider the Lovelace-Shapiro amplitude, discussed in more detail in 2203.11950. This amplitude satisfies 1SDR, t-channel dominance, and crossing. T-channel dominance can be thought of as what happens at large N in 2203.11950, so small but non-zero rho_2 might be relevant for finite but large N. I don’t think anyone has looked at that.

That last statement near 5.24 is just wrong, I’m not sure what it was supposed to say but we’ve removed it.

Open ended question: we have thought about this but we don’t see how to make this argument. T-channel dominance really seems to be related to the large N limit for pions. I suppose you mean to try to show that having e^2 < G requires rho_2 = 0? I don’t see a way to prove this analytically this work but it would be interesting to try numerically. For instance, one could perhaps fix G and e^2 and then maximize rho_2 at different values. We added a few sentences about this as a possible future direction in the discussion on page 38.

End of page 13 – fixed (should have said 5.2).

Caption of figure 11: we’ve added a brief explanation in the caption

---

## Editorial Decision

in_voting